# RuCL: Stratified Rubric-Based Curriculum Learning for Multimodal Large Language Model Reasoning

**Yukun Chen** [* 1 2]  **Jiaming Li** [* 1 2]  **Longze Chen** [1 2]  **Ze Gong** [† 1]  **Jingpeng Li** [3]  **Zhen Qin** [3]  **Hengyu Chang** [3]
**Lei Zhang** [1 2]  **Ancheng Xu** [1 2]  **Zhihao Yang** [2]  **Hamid Alinejad-Rokny** [4]  **Qiang Qu** [1]  **Bo Zheng** [† 3]  **Min Yang** [† 1 5]

## Abstract

Reinforcement Learning with Verifiable Rewards (RLVR) has emerged as a prevailing paradigm for enhancing reasoning in Multimodal Large Language Models (MLLMs). However, relying solely on outcome supervision risks reward hacking, where models learn spurious reasoning patterns to satisfy final answer checks. While recent rubric-based approaches offer fine-grained supervision signals, they suffer from high computational costs of instance-level generation and inefficient training dynamics caused by treating all rubrics as equally learnable. In this paper, we propose **Stratified Rubric-Based Curriculum Learning (RuCL)**, a novel framework that reformulates curriculum learning by shifting the focus from data selection to reward design. RuCL generates generalized rubrics for broad applicability and stratifies them based on model competence, dynamically adjusting their weights to guide the model from foundational perception to advanced logical reasoning. Extensive experiments on various visual reasoning benchmarks show that RuCL yields a remarkable **+7.83%** average improvement over the Qwen2.5-VL-7B model, achieving a state-of-the-art accuracy of **60.06%**.

## 1. Introduction

Multimodal Large Language Models (MLLMs) have demonstrated remarkable capabilities in complex visual reasoning tasks, spanning from mathematical problem-solving to chart

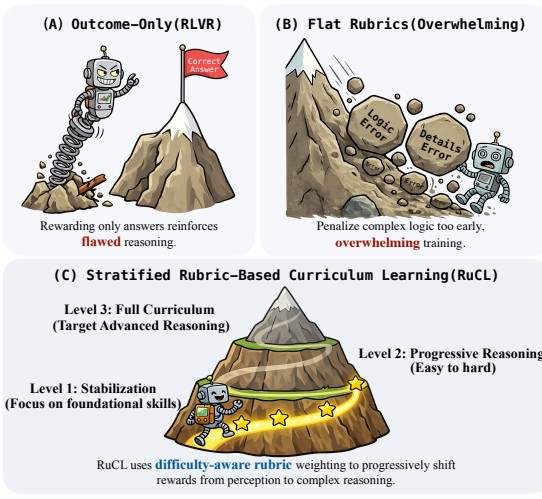

*Figure 1.* **Comparison of reward paradigms.** We move beyond (A) outcome-only signals and (B) unstructured dense feedback. (C) Our **RuCL** framework organizes rubrics into a stratified curriculum, aligning reward complexity with the model's progressive learning stages.

understanding (Yao et al., 2024; Liu et al., 2025b; Peng et al., 2025; Amizadeh et al., 2020; Garcez et al., 2019). To further augment these reasoning capabilities, Reinforcement Learning with Verifiable Rewards (RLVR) (Shao et al., 2024; Cui et al., 2025; Li et al., 2025) has emerged as a prevalent post-training paradigm. By employing straightforward rule-based verification, RLVR avoids the reliance on costly reward models (Meng et al., 2025; Liu et al., 2025a; Xu et al., 2025b). However, this outcome-based reward mechanism suffers from a fundamental limitation: it overemphasizes final answer correctness at the expense of intermediate reasoning quality. As a result, models are prone to learning spurious reasoning patterns or exploiting superficial shortcuts. This frequently leads to the generation of contradictory or hallucinatory intermediate steps that serendipitously arrive at correct answers. This "reward hacking" phenomenon severely compromises the reliability of the reasoning.

While recent LLM-as-a-Judge frameworks successfully mitigate reward hacking by constructing rubrics to assess the

---

[*]Equal contribution    [†]Corresponding authors. [1]Shenzhen Institutes of Advanced Technology, Chinese Academy of Sciences [2]University of Chinese Academy of Sciences [3]Alibaba Group [4]School of Biomedical Engineering, UNSW Sydney [5]Shenzhen University of Advanced Technology. Correspondence to: Ze Gong <ze.gong@siat.ac.cn>, Bo Zheng <bozheng@alibaba-inc.com>, Min Yang <min.yang@siat.ac.cn>.

*Proceedings of the $43^{rd}$ International Conference on Machine Learning*, Seoul, South Korea. PMLR 306, 2026. Copyright 2026 by the author(s).

validity of reasoning trajectories (Viswanathan et al., 2025; Gunjal et al., 2025), they are hampered by two fundamental limitations (Huang et al., 2025b; Zhou et al., 2025; Pathak et al., 2025). First, generating rubrics at the *instance level* incurs high computational overhead, especially during an online reinforcement learning setting. Second, and more importantly, existing methods treat all rubrics as equally challenging throughout the training process, lacking a principled mechanism to account for heterogeneous learnability across evaluation rubrics. Consequently, models are penalized for complex logical failures before mastering basic skills such as visual perception, resulting in noisy gradient signals and hindering efficient convergence.

Drawing inspiration from Curriculum Learning (CL) (Bengio et al., 2009; Parashar et al., 2025), which traditionally organizes training data from easy to hard, we propose **Stratified Rubric-Based Curriculum Learning (RuCL)**, a novel framework that applies curriculum learning directly to reward design rather than data selection. Instead of treating all rubrics uniformly throughout training, our key insight is to organize and schedule rubrics according to their learnability, enabling the model to acquire reasoning skills in a structured and progressive manner (Fig. 1).

RuCL can be explained as a two-phase process: **(1) Generalized Rubric Construction and Stratification**: We adopt a data-driven approach to generate generalized rubrics that capture essential reasoning primitives shared across tasks, rather than relying on costly instance-specific evaluation. We estimate the model's initial competence on each rubric and stratify them by empirical proficiency level, ranging from foundational skills to advanced reasoning abilities. **(2) Dynamic Curriculum Learning**: During training, RuCL dynamically adjusts the weights of these rubrics based on the model's evolving capabilities. Training initially prioritizes foundational rubrics (e.g., visual element recognition). As the model demonstrates competence, the framework automatically shifts focus towards hard rubrics (e.g., complex logical deduction), effectively guiding the model from basic perception to advanced reasoning. Finally, the combination of final answer reward and rubric-based reward jointly promotes the model's reasoning capabilities.

Our contributions are summarized as follows:

(i) We introduce RuCL, a reward-centric curriculum framework that dynamically aligns rubric difficulty with model competence.

(ii) We instantiate RuCL with a data-driven rubric construction pipeline, an applicability-aware evaluation mechanism, and a performance-triggered curriculum scheduler, yielding a practical and scalable reward design for rubric-based approaches.

(iii) We conduct extensive experiments across seven benchmarks, showing that RuCL achieves an average performance gain of **7.83%**, and provide detailed ablation studies validating its effectiveness.

**Conflict of Interest Disclosure**  The authors J. Li, Z. Qin, H. Chang, and B. Zheng are employed by Alibaba Group, which leads the development of Qwen models, which were among the ones evaluated in this paper.

## 2. Related Work

**Post-training for MLLMs.**  Early MLLM reasoning methods, such as LLaVA-Reasoner (Zhang et al., 2025), MPO (Wang et al., 2024b), and Insight-V (Rafailov et al., 2023), rely on rationale distillation, human preferences, or iterative DPO, but are limited by heavy supervision and low scalability. To address this, Reinforcement Learning with Verifiable Rewards (RLVR) (Ma et al., 2025; Chu et al., 2025) verifies final answers against ground truth, enabling scalable reasoning improvement. For example, Vision-R1 (Huang et al., 2025a) leverages teacher MLLMs to generate chain-of-thought (CoT) data, DeepScaler (Luo et al., 2025) and Light-R1 (Wen et al., 2025) combine supervised and RL training, and VL-Rethinker (Wang et al., 2025a), SRPO (Wan et al., 2025), and GThinker (Zhan et al., 2025) use reflection-aware rewards. Recent multi-turn RL methods further optimize iterative refinement and self-correction in code-generation settings, either through single-step rewards over multi-turn trajectories (Jain et al., 2025) or multi-turn GRPO (Ekbote et al., 2025). These works focus on interaction structure and self-correction, while RuCL targets the reward structure itself. Despite these advances, sparse outcome-based rewards leave models prone to reward hacking via spurious reasoning.

**Rubrics as Rewards.**  To address the opacity and sparsity of outcome-based supervision, recent work uses structured rubrics to evaluate intermediate reasoning processes, decomposing tasks into explicit, verifiable criteria. Rubrics have proven effective in domains such as medical reasoning (Arora et al., 2025), code generation (Mahdaoui et al., 2025), and instruction following (Pathak et al., 2025; Galvan-Sosa et al., 2025; Fan et al., 2024; Winata et al., 2025). LLM-as-a-Judge frameworks (Team et al., 2025a; Viswanathan et al., 2025) integrate rubrics into reinforcement learning, providing more informative reward signals than standard RLVR (Huang et al., 2025b; Gunjal et al., 2025). However, existing approaches typically generate instance-specific rubrics and treat all rubrics as equally learnable, lacking a principled mechanism to account for heterogeneous difficulty across reasoning skills. Besides, recent preference-based reward modeling studies richer ordinal reward signals learned from preferences (Chen et al., 2026). In contrast, RuCL keeps rubric evaluation explicit and interpretable, and focuses on how to stratify and schedule

multiple rubric-level rewards during RL.

**Curriculum Learning.** Curriculum Learning (CL), introduced by (Bengio et al., 2009), organizes training into phases to mimic human learning and enable progressive skill acquisition (Parashar et al., 2025; Shi et al., 2025; Chen et al., 2025; Song et al., 2025). Kwai Keye-VL (Team et al., 2025b) improves capability and stability by adopting a multi-stage training recipe that structures both pre-training and post-training, while VL-Contigo (Yuan et al., 2025) implements an "easy-to-hard" RL curriculum with online difficulty weighting across three stages. Progressive reward shaping also adapts reward signals over optimization to improve exploration and stability (Su et al., 2025). These prior approaches focus on data-level curricula or hand-crafted reward-shaping stages; in contrast, we apply CL at the rubric level, dynamically adjusting rubric weights during RL to balance training stability and reasoning performance.

# 3. Stratified Rubric-Based Curriculum Learning (RuCL)

In this work, we focus on rubric-based rewards to improve the reasoning capabilities of Multimodal Large Language Models (MLLMs). While rubrics provide fine-grained supervision over reasoning processes, existing methods typically combine them with fixed weights, ignoring differences in difficulty and learnability. This results in noisy gradients and inefficient optimization. We propose **Stratified Rubric-Based Curriculum Learning (RuCL)**, which applies curriculum learning directly to reward design by progressively emphasizing rubrics of increasing difficulty. In this section, we first formalize the learning objective, and then detail rubric construction and the curriculum mechanism.

## 3.1. Problem Formulation

We consider a Reinforcement Learning (RL) setting. Given an input query $x$ (e.g., an image-text pair), a policy $\pi_\theta(y \mid x)$ generates a response $y$. The learning signal is provided by a scalar reward function $r^{(t)}(y \mid x)$, which may vary over the training steps $t$ to reflect the dynamic curriculum scheduling of supervision signals. This reward integrates multiple sources of supervision, including (i) *rule-based* verification of final answer correctness, and (ii) *rubric-based* evaluations that assess intermediate reasoning qualities such as perception, grounding, and logical consistency. These rubric signals are derived from a set of evaluation rubrics $\mathcal{R} = \{R_1, \ldots, R_k\}$, each targeting a distinct reasoning aspect. Our objective is to learn a policy $\pi_\theta$ that maximizes the expected reward:

$$\max_\theta \ \mathbb{E}_{x \sim \mathcal{D}, \, y \sim \pi_\theta(\cdot \mid x)} \left[ r^{(t)}(y \mid x) \right]. \quad (1)$$

We optimize this objective using Group Relative Policy Optimization (GRPO) (Shao et al., 2024), a stable policy-gradient method for RLVR (see Appendix A for details). The core challenge is to design $r^{(t)}$ such that it provides adaptive supervision across reasoning skills that differ substantially in difficulty and learnability.

**Rubric Rewards as Multi-Objective Optimization.** Rubric-based supervision can be viewed as optimizing multiple skill-wise objectives under a shared policy. Specifically, each rubric $R_k \in \mathcal{R}$ induces a sub-reward $r_k(y \mid x)$, and the overall rubric reward corresponds to a weighted combination:

$$r_{\mathrm{rub}}^{(t)}(y \mid x) = \sum_{k=1}^{K} \omega_k^{(t)} \, r_k(y \mid x), \quad \text{s.t.} \sum \omega_k^{(t)} = 1. \quad (2)$$

A key challenge in this formulation is the heterogeneity of the objectives. The rubrics range from basic checks to complex reasoning steps, implying that their corresponding reward signals vary significantly in density and reliability. Indiscriminately mixing these diverse signals with static weights risks letting noisy, high-difficulty objectives dominate or interfere with the learning of foundational skills. Therefore, a time-varying weighting scheme $\omega^{(t)}$ naturally serves as a curriculum over reward components, allowing optimization to prioritize learnable, low-noise objectives first and progressively incorporate harder reasoning criteria.

In the following sections, we detail how $r^{(t)}$ is instantiated via data-driven rubric construction and difficulty stratification (Sec. 3.2), and how the curriculum is scheduled via a performance-triggered mechanism (Sec. 3.3). The overview of RuCL is illustrated in Fig. 2.

## 3.2. Phase I: Generalized Rubric Construction and Stratification

To construct a robust and discriminative reward system, we design a quantitative, data-driven pipeline that filters and stratifies rubrics based on their empirical behavior. In contrast to existing rubric-based methods that generate ad hoc, instance-specific rubrics (Zhou et al., 2025; Gunjal et al., 2025), we construct a reusable set of generalized rubrics that remain applicable across diverse reasoning tasks, enabling principled difficulty stratification and curriculum scheduling.

**Computational Efficiency Analysis.** We distinguish the cost of RuCL from instance-specific rubric pipelines (Jia et al., 2025; Zhou et al., 2025; Gunjal et al., 2025) along two axes: rubric generation and reward evaluation. Let $N$ denote the number of unique training queries, $C_{\mathrm{gen}}$ the cost of generating a rubric set, and $C_{\mathrm{eval}}$ the cost of evaluating a response with a judge. Instance-specific methods synthesize a tailored rubric set for each query, incurring a

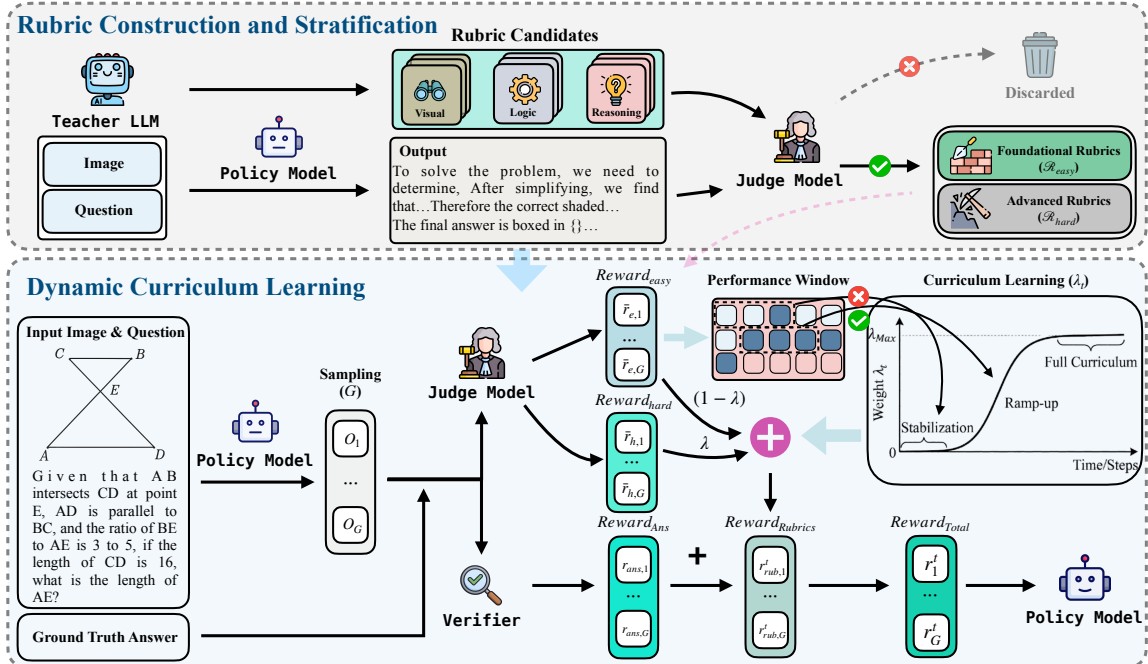

*Figure 2*. **Overview of Stratified Rubric-Based Curriculum Learning (RuCL).** The framework proceeds in two stages: (Top) **Generalized Rubric Construction and Stratification**, where evaluation rubrics are generated and categorized into Foundational ($\mathcal{R}_{\text{easy}}$) and Advanced ($\mathcal{R}_{\text{hard}}$) tiers based on empirical difficulty. (Bottom) **Dynamic Curriculum Learning**, where the rubric-based reward is synthesized via a dynamic weighting mechanism controlled by a scheduler. By adjusting the weight $\lambda$ based on real-time performance, RuCL progressively shifts the optimization focus from mastering basic skills to tackling complex reasoning.

generation cost of $\mathcal{O}(NC_{\text{gen}})$ before or during training. In contrast, RuCL constructs a single generalized rubric pool offline and reuses it for all queries, reducing the generation cost to $\mathcal{O}(C_{\text{gen}})$. Both paradigms may still use a strong LLM judge during RL, so the evaluation cost remains a nontrivial training-time expense. However, this cost is incurred only for offline policy optimization and does not affect deployment-time inference, where the trained policy is used without the teacher or judge. This separation yields a favorable amortized trade-off: RuCL retains dense rubric-level supervision during training while avoiding repeated per-instance rubric generation and persistent inference-time overhead.

**Candidate Generation & Rollout.** We prompt a teacher LLM with comprehensive context, including the task category, relevant images, the input query, and the ground truth answer, and instruct it to generate a diverse set of the most relevant rubric candidates ($\mathcal{R}_{\text{candidates}}$). We then perform rollouts on a randomly sampled subset of training instances ($\mathcal{D}_{\text{sample}}$) of size $N$ using the base model to collect rubric-level evaluation signals.

**Applicability-Aware Evaluation.** Unlike standard scalar scoring, we design a specialized Judge mechanism that explicitly decouples *relevance* from *performance*. For each sample $x_i \in \mathcal{D}_{\text{sample}}$ and rubric candidate $R_j$, the Judge out-

puts a tuple $(a_{ij}, s_{ij})$, where $a_{ij} \in \{0, 1\}$ indicates whether rubric $R_j$ is applicable to the problem context of $x_i$, and $s_{ij} \in \{0, 1\}$ denotes whether the model output satisfies the rubric, evaluated only when $a_{ij} = 1$.

This explicit decoupling ensures that the computed statistics accurately reflect the rubric's effective coverage and the model's actual proficiency. By preventing non-applicable rubrics from skewing the metrics, this mechanism provides a reliable basis for selecting high-coverage rubrics and stratifying them by difficulty. The detailed evaluation prompt is provided in Appendix G.

**Metric-Based Filtering and Stratification.** Using the assessment statistics, we refine the candidate pool to construct a structured curriculum. We first compute the **Applicability Rate ($\eta_j$)** to quantify each rubric's coverage across the dataset:

$$\eta_j = \frac{1}{N} \sum_{i=1}^{N} a_{ij}. \qquad (3)$$

To ensure broad coverage and reduce noise from rarely applicable rubrics, we discard rubrics with insufficient coverage ($\eta_j < \tau_{\text{app}}$). We provide detailed statistics in Appendix F.2 illustrating the high variance in rubric coverage (e.g., as low as 9.7%), which empirically justifies the necessity of this filtering mechanism to avoid catastrophic gradient noise.

*Table 1.* **The stratified reward system.** The evaluation rubrics are categorized by difficulty and implemented via either a generative LLM Judge or a deterministic Answer Verifier. Detailed rubric definitions and scoring criteria are provided in Appendix F.

| Rubric Criterion | Evaluation Focus | Difficulty Stratum | Evaluator |
|---|---|---|---|
| VISUAL PRESENCE | Penalizes object and attribute hallucinations. | | |
| ENTITY EXTRACTION | Isolates the specific Region of Interest (ROI). | Foundational ($\mathcal{R}_{\text{easy}}$) | |
| INTENT ALIGNMENT | Checks compliance with scope and constraints. | | LLM Judge |
| CONCLUSION MATCH | Ensures the answer logically follows the reasoning. | | |
| STEP COHERENCE | Detects logical gaps and internal contradictions. | Advanced ($\mathcal{R}_{\text{hard}}$) | |
| EVIDENCE GROUNDING | Validates inferences using specific visual cues. | | |
| ANSWER CORRECTNESS | Verifies the final answer against the ground truth. | — | Answer Verifier |

For the remaining rubrics ($\mathcal{R}_{\text{filtered}} \subseteq \mathcal{R}_{\text{candidates}}$), we compute the **Pass Rate** ($p_j$), defined as the current model's conditional success rate on applicable instances:

$$p_j = \frac{\sum_{i=1}^{N}(a_{ij} \cdot s_{ij})}{\sum_{i=1}^{N} a_{ij}}. \tag{4}$$

This metric serves as an empirical proxy for difficulty, allowing us to stratify rubrics based on their role in the learning process. We partition them into two distinct levels (see Table 1): **a) Foundational Rubrics ($\mathcal{R}_{\text{easy}}$)**, characterized by high pass rates, target prerequisite skills to provide stable initial supervision signals; **b) Advanced Rubrics ($\mathcal{R}_{\text{hard}}$)**, identified by low pass rates, target complex reasoning gaps that remain underdeveloped in the base model. This separation enables a curriculum that reinforces basics first, then progressively pivots to challenging reasoning tasks.

**Statistical Interpretation of Pass Rate as Difficulty Proxy.** We justify using the pass rate as a principled indicator of optimization difficulty through the lens of gradient estimator stability. For a fixed rubric $R_j$, we model its signal as a Bernoulli variable $r_j \sim \text{Bernoulli}(p_j)$. In policy gradient methods, the reliability of the update is inversely related to the Coefficient of Variation (CV) of the estimator:

$$CV(r_j) = \frac{\sqrt{Var(r_j)}}{\mathbb{E}[r_j]} = \frac{\sqrt{p_j(1-p_j)}}{p_j} = \sqrt{\frac{1}{p_j} - 1}. \tag{5}$$

This derivation reveals a critical insight: as the pass rate $p_j \to 0$, the relative noise diverges ($CV \to \infty$). This implies that rubrics with low pass rates (Advanced Rubrics) provide gradient signals that are dominated by noise, leading to inefficient credit assignment. Conversely, high-pass-rate rubrics (Foundational Rubrics) offer low-CV, reliable signals. Thus, stratifying rubrics by pass rate is statistically equivalent to stratifying by gradient reliability.

### 3.3. Phase II: Dynamic Curriculum Learning

We employ a hybrid reward mechanism that integrates rule-based correctness with the stratified rubrics derived in Phase

I. We introduce a stability-aware curriculum that dynamically adjusts the focus from foundational to advanced reasoning.

**Hybrid Reward Components.** Our reward system adopts a hybrid evaluation strategy that integrates model-based rubric evaluation with strict rule-based verification, balancing fine-grained rubric-level process supervision with unambiguous outcome correctness. We employ a strict rule-based verifier to assess the final answer correctness. For each sampled response $y_i$ conditioned on input $x$, the final outcome reward is defined as:

$$r_{\text{ans}}(y_i \mid x) = \mathbb{I}(\text{grade}(\hat{y}_i, y^*) = 1), \tag{6}$$

where $\hat{y}_i$ is the extracted prediction and $y^*$ is the ground truth.

In parallel, we evaluate the reasoning process using the foundational ($\mathcal{R}_{\text{easy}}$) and advanced ($\mathcal{R}_{\text{hard}}$) rubric sets derived in Sec. 3.2. During training, the Judge model evaluates the generated response against all rubrics in these filtered sets. We aggregate the binary satisfaction signals to compute tier-level reasoning scores:

$$\begin{aligned} \bar{r}_{\text{easy}}(y_i \mid x) &= \frac{1}{|\mathcal{R}_{\text{easy}}|} \sum_{R \in \mathcal{R}_{\text{easy}}} r(y_i \mid x, R), \\ \bar{r}_{\text{hard}}(y_i \mid x) &= \frac{1}{|\mathcal{R}_{\text{hard}}|} \sum_{R \in \mathcal{R}_{\text{hard}}} r(y_i \mid x, R), \end{aligned} \tag{7}$$

where $r(y_i \mid x, R) \in \{0, 1\}$ denotes whether the response satisfies rubric $R$. These aggregated scores $\bar{r}_{\text{easy}}$ and $\bar{r}_{\text{hard}}$ serve as the basis for our curriculum scheduling mechanism.

**Performance-Triggered Curriculum Scheduling.** We introduce a **Stability-Aware Curriculum** that regulates the progression from foundational to advanced reasoning supervision. In contrast to static schedules, RuCL activates advanced rubrics only after the model demonstrates stable proficiency on foundational ones.

For a sampled response $y_i$ at training step $t$, we define the

*Table 2.* Performance comparison on Mathematical Reasoning and General benchmarks. The "Avg." column reports the average score across all seven evaluated benchmarks. Green deltas denote absolute gains over the base model. The best results among open-source reasoning models are highlighted in **bold**, while the second-best are underlined.

| Model | Mathematical Reasoning | | | | General | | | Avg. |
|---|---|---|---|---|---|---|---|---|
| | MathVerse | MathVision | MathVista | WeMATH | MMMU | LogicVista | Counting | |
| **Proprietary Models** | | | | | | | | |
| GPT-4o | 50.20 | 30.30 | 63.80 | 68.80 | 69.10 | 45.90 | - | - |
| Claude-3.5-Sonnet | 57.64 | 46.48 | 67.70 | 73.05 | 68.30 | 43.97 | - | - |
| **Open-Source General-Purpose Models** | | | | | | | | |
| Qwen2.5-VL-7B | 48.98 | 24.18 | 70.20 | 58.52 | 51.00 | 39.26 | 73.50 | 52.23 |
| Qwen2.5-VL-32B | 57.60 | 38.40 | 74.70 | 69.10 | 57.44 | 49.26 | 85.36 | 61.69 |
| InternVL2.5-8B | 39.53 | 19.70 | 62.30 | 53.50 | 45.73 | 38.23 | 74.83 | 47.69 |
| InternVL2.5-38B | 49.40 | 32.20 | 71.84 | 68.61 | 56.98 | 47.21 | 82.77 | 58.43 |
| **Open-Source Multimodal Large Language Models** | | | | | | | | |
| MM-Eureka-7B | 51.09 | 27.70 | 73.00 | 65.34 | 53.78 | 47.87 | 75.50 | 56.33 |
| OpenVLThinker-7B | 48.37 | 25.90 | 71.38 | 66.63 | 54.29 | 36.24 | 65.00 | 52.54 |
| Perception-R1-7B | 52.56 | 28.06 | 72.80 | 65.57 | 53.11 | 40.94 | 82.30 | 56.48 |
| Vision-R1-7B | 53.23 | 27.24 | 70.63 | 64.98 | 43.28 | 42.94 | 83.27 | 55.08 |
| R1-Onevision-7B | 45.12 | 23.91 | 66.21 | 61.88 | 43.70 | 44.53 | 78.45 | 51.97 |
| ThinkLite-VL-7B | 51.47 | 27.24 | 73.30 | 65.52 | 55.44 | 42.94 | **86.50** | 57.49 |
| VL-Rethinker-7B | 53.86 | **29.57** | 73.27 | 68.22 | 54.67 | 46.08 | 68.50 | 56.31 |
| **RuCL** | **54.14** | 28.88 | **74.10** | **71.49** | **56.67** | **49.66** | 85.50 | **60.06** |
| Δ(*Qwen2.5-VL-7B*) | ↑ *5.16* | ↑ *4.70* | ↑ *3.90* | ↑ *12.97* | ↑ *5.67* | ↑ *10.40* | ↑ *12.00* | ↑ *7.83* |

curriculum-modulated rubric reward as:

$$r_{\text{rub}}^{(t)}(y_i \mid x) = (1-\lambda_t) \cdot \bar{r}_{\text{easy}}(y_i \mid x) + \lambda_t \cdot \bar{r}_{\text{hard}}(y_i \mid x), \quad (8)$$

where the curriculum coefficient $\lambda_t \in [0, \lambda_{\max}]$ controls the difficulty mix between foundational and advanced reasoning rubrics. Initially, $\lambda_t$ is set to zero and remains unchanged until foundational performance stabilizes.

The curriculum proceeds in three phases:

**(1) Stabilization Phase:** We enforce $\lambda_t = 0$. Let $\mu_{\text{easy}}^{(t)} = \mathbb{E}_{(x,y)\sim\mathcal{B}_t}[\bar{r}_{\text{easy}}(y \mid x)]$ denote the batch-averaged foundational rewards at step $t$, and let $W_t = \{\mu_{\text{easy}}^{(t-w+1)}, \ldots, \mu_{\text{easy}}^{(t)}\}$ be a sliding window of length $w$. The transition is triggered at step $T_{\text{start}}$ only when the model's performance consistently exceeds a proficiency threshold $\tau_{th}$ throughout the entire window:

$$T_{\text{start}} = \min\{t \mid \forall \mu \in W_t, \mu \geq \tau_{th}\}. \quad (9)$$

This strict condition ensures that the model does not progress to advanced reasoning stages due to transient lucky guesses.

**(2) Curriculum Ramp-up:** Once triggered ($t > T_{\text{start}}$), $\lambda_t$ follows a defined growth function (e.g., Linear or Sigmoid) over a duration $T_{\text{ramp}}$:

$$\lambda_t = \lambda_{\text{base}} + (\lambda_{\max} - \lambda_{\text{base}}) \cdot \phi\left(\frac{t - T_{\text{start}}}{T_{\text{ramp}}}\right), \quad (10)$$

where $\phi(\cdot)$ is the normalized growth function clamped to $[0, 1]$ and $\lambda_{\text{base}}$ denotes the initial curriculum weight.

**(3) Advanced Consolidation:** Upon completion of the ramp-up period ($t > T_{\text{start}} + T_{\text{ramp}}$), the curriculum holds the difficulty weight at its peak: $\lambda_t = \lambda_{\max}$. We combine the rule-based outcome reward with the curriculum-modulated rubrics reward to obtain the scalar reward used by GRPO:

$$r^{(t)}(y_i \mid x) = \alpha \cdot r_{\text{ans}}(y_i \mid x) + (1-\alpha) \cdot r_{\text{rub}}^{(t)}(y_i \mid x). \quad (11)$$

Here, $r^{(t)}(y_i \mid x)$ is used in advantage estimation. We treat $\alpha \in [0, 1]$ as a fixed hyperparameter that controls the trade-off between outcome correctness and rubric-based process supervision.

**Effect of Curriculum on Gradient Variance and Optimization Stability.** We analyze how the curriculum coefficient $\lambda_t$ affects the stability of policy optimization. Recall that the curriculum-modulated rubric reward is a convex combination of two stochastic signals as defined in Eq. 8. Under policy-gradient learning, the gradient estimator is proportional to $\nabla_\theta \log \pi_\theta(y \mid x) \, r_{\text{rub}}^{(t)}$, whose variance can be expressed as

$$\text{Var}(\nabla J_t) = (1-\lambda_t)^2 \text{Var}_e + \lambda_t^2 \text{Var}_h + 2\lambda_t(1-\lambda_t)\text{Cov}_{e,h}, \quad (12)$$

where $\text{Var}_e$ and $\text{Var}_h$ denote the gradient variances induced by $\bar{r}_{\text{easy}}$ and $\bar{r}_{\text{hard}}$, respectively, and $\text{Cov}_{e,h}$ denotes their covariance.

In early training stages, advanced rubrics typically have low pass rates and sparse activations, making $\text{Var}_h$ orders of magnitude larger than $\text{Var}_e$. Meanwhile, $\text{Cov}_{e,h}$ is bounded by $\sqrt{\text{Var}_e \text{Var}_h}$ and empirically close to zero due to the near-orthogonality of the two reward sources. Under the regime $\text{Var}_h \gg \text{Var}_e$ and $|\text{Cov}_{e,h}| \ll \text{Var}_h$, the unconstrained minimizer is

$$\lambda^* = \frac{\text{Var}_e - \text{Cov}_{e,h}}{\text{Var}_e + \text{Var}_h - 2\text{Cov}_{e,h}} \approx \frac{\text{Var}_e - \text{Cov}_{e,h}}{\text{Var}_h},$$

which lies close to zero when $\text{Cov}_{e,h}$ is not comparable to $\text{Var}_e$. Thus, small values of $\lambda_t$ are variance-efficient in early training. As training progresses, the effective variance of hard-rubric gradients decreases, making larger values of $\lambda_t$ less destabilizing.

More broadly, while traditional curriculum learning re-shapes the *input distribution* by ordering examples from easy to hard, our method modulates the density of evaluative signals over the output space. When $\lambda_t$ is small, the policy receives substantive feedback only from foundational rubric dimensions, restricting early updates to a subspace with reliable learning signals—harder rubrics contribute near-zero or high-variance signals and are effectively masked. As $\lambda_t$ increases, this subspace gradually expands, admitting harder objectives only once their gradients become informative. The variance decomposition is derived in Appendix B.

## 4. Experiments

### 4.1. Experiment Setup

**Datasets & Models.** In our experiments, we utilize the ViRL-39K dataset (Wang et al., 2025a) for model training. ViRL-39K is a large-scale, high-quality dataset specifically curated for vision-language reinforcement learning (RL). It comprises approximately 39,000 verifiable question-answering pairs that cover a wide range of complex scenarios, including STEM, spatial reasoning, and multi-disciplinary chart analysis. Specifically, we initialize our training from Qwen2.5-VL-7B-Instruct (Bai et al., 2025) as the base model, leveraging its advanced multi-modal perception and robust instruction-following capabilities to facilitate further reasoning-oriented optimization.

**Evaluation.** We evaluate RuCL on widely used visual reasoning benchmarks covering multimodal mathematical reasoning and general visual reasoning. For multimodal mathematical reasoning, we use MathVista (Lu et al., 2023), MathVerse (Zhang et al., 2024), MATH-Vision (Wang et al., 2024a), and WeMATH (Qiao et al., 2025). For general visual reasoning, we employ LogicVista (Xiao et al., 2024), Super-CLEVR Counting (Li et al., 2023), and MMMU (Yue et al., 2024) to assess logical deduction, compositional counting and perception, and multi-disciplinary knowledge, respectively.

**Baselines.** We compare our model with several strong MLLMs, categorized into three groups: (1) *Proprietary models*, including GPT-4o (Hurst et al., 2024) and Claude-3.5-Sonnet (Anthropic, 2024); (2) *Open-source general-purpose models*, such as Qwen2.5-VL-7B-Instruct, Qwen2.5-VL-32B-Instruct (Bai et al., 2025), InternVL2.5-8B and InternVL2.5-38B (Chen et al., 2024); and (3) *Open-source reasoning-focused models*, including MM-Eureka-7B (Meng et al., 2025), OpenVLThinker-7B (Deng et al., 2025), Perception-R1-7B (Xiao et al., 2025), Vision-R1-7B (Huang et al., 2025a), R1-Onevision-7B (Yang et al., 2025), ThinkLite-VL-7B (Wang et al., 2025b), and VL-Rethinker-7B (Wang et al., 2025a). More information about the baselines is provided in Appendix D.

**Configuration.** For data-driven candidate generation, we utilize Gemini 3 Pro (Google DeepMind, 2025) as the teacher model. Rubric construction is performed once in an offline stage rather than repeatedly for each training instance during RL. Through few-shot prompting with representative context, we generate 20 rubric candidates. Subsequently, we conduct a rollout on $N = 2,000$ samples, retaining 6 core rubrics after filtering with an applicability threshold of 0.99. The retained rubrics are shared across all training instances and span both foundational perception-level checks and higher-level reasoning constraints. In the reinforcement learning phase, we deploy Qwen3-VL-235B-A22B-Instruct (Team, 2025) as the reward judge. The detailed prompts guiding the judge model's scoring process are provided in Appendix H. To implement the proposed *Stability-Aware Curriculum*, we configure the sliding window size $K = 20$ and the proficiency threshold $\tau_{th} = 0.9$. The reward balancing coefficient is set to $\alpha = 0.7$ to prioritize factual accuracy. All experiments are conducted on NVIDIA H200 GPUs using the `verl` framework (Sheng et al., 2025). Comprehensive hyperparameter details are provided in Appendix C.

### 4.2. Main Results

**Mathematical Reasoning Performance.** As shown in Table 2, RuCL demonstrates superior performance, outperforming the baseline Qwen2.5-VL-7B across all mathematical benchmarks. This significant improvement, driven by the integration of our fine-grained rubric-based reward modeling and curriculum learning strategy, validates the efficacy of prioritizing simple rubrics in early training stages before transitioning to harder reasoning constraints. Specifically, on the challenging WeMATH and MathVerse datasets, our model improves by 12.97% (from 58.52% to 71.49%) and 5.16% (from 48.98% to 54.14%), respectively. Furthermore, when compared with other leading open-source reasoning models such as ThinkLite-VL-7B and VL-Rethinker-7B, RuCL achieves the highest average score of 60.06% across all seven tasks, highlighting its robust reasoning capabilities.

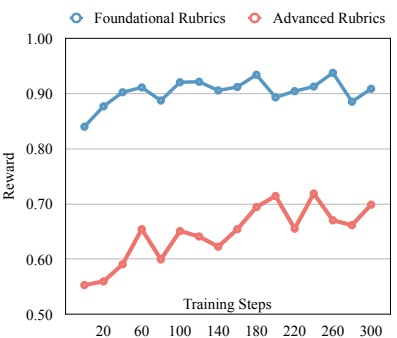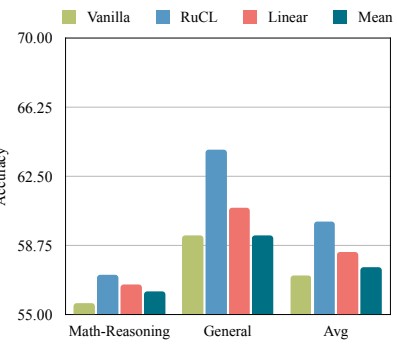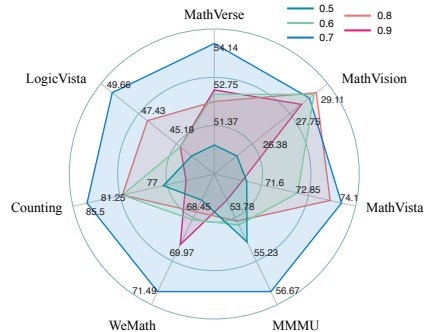

*Figure 3.* **Left:** Training dynamics of Foundational (blue) and Advanced (red) rubric rewards. **Middle:** Ablation study results on rubric aggregation and scheduling strategies. **Right:** Sensitivity analysis of the reward balancing hyperparameter.

*Table 3.* Sensitivity analysis of judge model choice. The top block reports standalone judge performance, and the bottom block reports policy performance after training with different judges.

| Model | MathVerse | MathVision | MathVista | MMMU |
|---|---|---|---|---|
| Judge 72B | 59.40 | 37.10 | 74.20 | 71.00 |
| Judge 235B | 72.50 | 59.00 | 84.90 | 78.70 |
| Base | 48.98 | 24.18 | 70.20 | 51.00 |
| RuCL(72B Judge) | 53.64 | 28.60 | 73.69 | 56.33 |
| RuCL(235B Judge) | **54.14** | **28.88** | **74.10** | **56.67** |

We further compare RuCL with these two strongest open-source baselines under multiple random seeds and evaluate RuCL on additional Qwen3-VL model scales. The results are provided in Appendix E.

**Generalization to General and Logical Benchmarks.** Extending beyond mathematics, our model achieves competitive results across broader reasoning tasks. As shown in the *General* section of Table 2, RuCL exhibits remarkable generalization. On the LogicVista benchmark, which requires complex logical deduction, our model achieves a 10.40% improvement over the baseline (from 39.26% to 49.66%), surpassing all other open-source 7B competitors. Similarly, we observe substantial gains on the comprehensive MMMU (+5.67%) and Counting (+12.00%) benchmarks, with the latter highlighting enhanced fine-grained visual perception (85.50%). These results indicate that combining intermediate rubric rewards with final outcome supervision effectively enhances the model's fundamental reasoning robustness rather than merely overfitting to mathematical domains. Notably, despite its compact scale, RuCL significantly narrows the performance gap with top-tier proprietary models.

**Training Dynamics and Curriculum Efficacy.** To validate the efficacy of RuCL, we analyze the evolution of reward trajectories throughout the training process, as shown in Figure 3. Initially, the curriculum prioritizes foundational rubrics, leading to the rapid mastery of prerequisite skills such as visual presence and entity extraction. As the mech-

anism detects stable proficiency (scores stabilizing $> 0.9$), it progressively introduces advanced reasoning constraints, and the model exhibits steady improvement in higher-order tasks while maintaining robust performance on foundational metrics. This demonstrates that RuCL fosters complex reasoning while preserving foundational visual perception and instruction-following skills. Although the difficulty of individual samples may change as the policy improves, RuCL operates on rubric-level stratification rather than instance-level difficulty. The stable foundational reward trajectory suggests that the curriculum is not disrupted by such sample-level changes. Furthermore, qualitative case studies in Appendix I provide concrete evidence of RuCL's capability to mitigate reward hacking. We show that our rubric-based judge effectively penalizes spurious reasoning chains that serendipitously arrive at the correct answer.

### 4.3. Ablation Study

In this section, we conduct ablation studies to validate the contributions of our key design choices. We focus on two key components: the rubric aggregation mechanism and the sensitivity to the reward balancing hyperparameter $\alpha$.

**Impact of Rubric Aggregation and Scheduling.** To assess the contribution of rubric aggregation and curriculum scheduling, we compare our method (**Sigmoid Stratification**) with the following baselines, keeping the GRPO backbone and training data fixed: **(1) Vanilla GRPO:** Trains using solely the rule-based outcome reward $r_{\text{ans}}$, ignoring all reasoning rubrics. **(2) Uniform Averaging:** Aggregates all filtered rubrics into a single unweighted average score, discarding difficulty stratification and curriculum scheduling. **(3) RuCL (Sigmoid Stratification):** Adopts the proposed stratified rubrics ($\mathcal{R}_{\text{easy}}, \mathcal{R}_{\text{hard}}$) with the stability-aware sigmoid schedule for $\lambda_t$. **(4) Linear Stratification:** Replaces the sigmoid growth function with a simple linear ramp for $\lambda_t$ to evaluate the impact of schedule shape.

Figure 3 highlights the aggregate trend: *Vanilla GRPO*

Table 4. Ablation results on General benchmarks.

| Method | MMMU | LogicVista | Counting | Avg. |
|---|---|---|---|---|
| Vanilla GRPO | 54.89 | 46.53 | 76.00 | 59.14 |
| Uniform Averaging | 55.44 | **50.11** | 77.00 | 59.29 |
| Linear Stratification | 55.44 | 47.43 | 79.50 | 60.79 |
| **RuCL** | **56.67** | 49.66 | **85.50** | **63.94** |

Table 5. Sensitivity analysis of sliding window size $w$ in curriculum triggering.

| Window Size $w$ | Mathematical | General | Avg. |
|---|---|---|---|
| $w = 10$ | 55.57 | 60.8 | 57.81 |
| $w = 20$ | **57.15** | **63.94** | **60.06** |
| $w = 30$ | 56.64 | 63.74 | 59.67 |

(57.13%) is surpassed by *Uniform Averaging* (57.56%) due to process supervision, while *Linear Stratification* (58.41%) yields further gains by distinguishing difficulty. As shown in Table 4, RuCL significantly outperforms the Linear strategy, particularly on perception-heavy tasks like Counting (85.50% vs. 79.50%) and logic-intensive tasks like LogicVista (49.66% vs. 47.43%). This advantage stems from the Sigmoid schedule's ability to reach maximum difficulty saturation earlier than the linear ramp. By completing the transition phase faster, Sigmoid affords the model a longer stable period to converge under the full weight of hard constraints, whereas the Linear approach keeps the reward signal in a continuous state of flux.

**Sensitivity to Judge Model Choice.** RuCL uses an LLM judge to provide rubric-level reward signals during training; we further examine whether the gains depend critically on a specific judge model. Table 3 first shows that Qwen3-VL-235B-A22B-Instruct is substantially stronger than Qwen2.5-VL-72B-Instruct on standalone benchmark performance, suggesting that the larger judge can provide higher-quality supervision. We then replace the 235B judge with the 72B judge while keeping all other training settings fixed. Both judge settings consistently improve over the base model, and the performance gap between RuCL(72B) and RuCL(235B) remains small across benchmarks. This indicates that stronger judges can be beneficial, but the effectiveness of RuCL is not solely attributable to a particular large judge; rather, the rubric-based curriculum design provides robust supervision across different judge choices.

**Sensitivity to Reward Balancing Hyperparameter $\alpha$.**

We further investigate the sensitivity of RuCL to the reward balancing coefficient $\alpha \in [0.5, 0.9]$, which controls the trade-off between outcome correctness and rubric-based process supervision in Eq. 11. In practice, we choose $\alpha$ through a limited sensitivity analysis over this range and report the trend to assess robustness, rather than introducing an additional validation-based model-selection stage. As shown in Figure 3 (right), performance improves as $\alpha$ increases from lower values, reaches the best overall result at $\alpha = 0.7$, and then declines when $\alpha$ becomes too large. This trend indicates that RuCL is moderately sensitive to the balance between outcome and process rewards. When $\alpha$ is too small, rubric-based process supervision dominates the objective and can distract optimization from final-answer

correctness, as reflected by the drop on Counting at $\alpha = 0.5$. Conversely, when $\alpha$ is too large, training approaches a sparse outcome-only reward regime, weakening the benefit of fine-grained rubric feedback and causing performance degradation on reasoning-heavy tasks such as WeMATH. Thus, $\alpha = 0.7$ provides the best empirical trade-off in our experiments, integrating intermediate reasoning guidance without overshadowing the primary objective of accurate problem solving.

**Sensitivity to Sliding Window Size $w$.** We study the sensitivity of the stability-aware trigger to the sliding window size $w$ in Eq. 9. We vary $w \in \{10, 20, 30\}$ while keeping all other hyperparameters fixed. As shown in Table 5, $w = 20$ achieves the best overall performance, while $w = 30$ performs comparably with only a marginal gap. In contrast, $w = 10$ consistently underperforms, indicating that a shorter window is more susceptible to transient fluctuations and may trigger the curriculum transition prematurely. Overall, these observations suggest that our curriculum mechanism is robust to moderate changes in $w$, and we adopt $w = 20$ as the default setting in all experiments.

## 5. Conclusion

We propose **Stratified Rubric-Based Curriculum Learning (RuCL)**, a framework that reframes curriculum learning from data selection to reward design. By stratifying evaluation rubrics into foundational and advanced categories, RuCL aligns reward signals with the model's evolving capabilities. Integrated with GRPO, this approach effectively mitigates reward hacking and training instability. Experiments across seven benchmarks demonstrate that RuCL significantly outperforms the base model and establishes a new state-of-the-art among 7B-scale reasoning models.

Several promising directions remain open for future investigation. Online rubric construction could eliminate reliance on offline teacher-generated rubrics by discovering evaluation criteria directly from training dynamics, enabling more task-specific and data-efficient reward design. Adaptive difficulty re-estimation that dynamically updates rubric stratification as the policy evolves may yield a more fine-grained curriculum. Extending RuCL to broader reasoning domains such as code generation could test the generality of rubric-based curriculum learning beyond visual tasks.

## Acknowledgements

Min Yang is supported by National Key Research and Development Program of China (2024YFF0908200), National Natural Science Foundation of China (Grant No. 62376262), the Natural Science Foundation of Guangdong Province of China (2024A1515030166, 2025B1515020032), and the Innovation Team Project of Guangdong Province (No. 2024KCXTD017). This work was supported by CCF-Alimama Tech Bag Fund.

## Impact Statement

This paper introduces Stratified Rubric-Based Curriculum Learning (RuCL), a framework that enhances the reasoning capabilities of Multimodal Large Language Models (MLLMs) by shifting curriculum focus from data selection to reward design. RuCL guides models to master foundational perception before progressing to advanced deduction, fostering the development of reliable models that prioritize intermediate reasoning integrity. RuCL utilizes widely recognized, publicly available datasets for training and evaluation, strictly adhering to their licenses and usage policies without intentionally introducing private, personally identifiable information (PII) or offensive content, ensuring that our advancements in multimodal intelligence are built upon transparent and reproducible foundations.

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

## A. Group Relative Policy Optimization (GRPO)

To enhance the reasoning capabilities of our model, we optimize the policy $\pi_\theta$ using Group Relative Policy Optimization (GRPO) (Shao et al., 2024). Unlike standard Proximal Policy Optimization (PPO) (Schulman et al., 2017), which necessitates a separate value function (critic) for advantage estimation, GRPO reduces computational overhead by leveraging group-based statistics. Specifically, for each input query $x$, we sample a group of $G$ outputs $\{y_i\}_{i=1}^{G}$ from $\pi_{\theta_{\text{old}}}$. The advantage $\hat{A}_i$ for the $i$-th output is estimated by normalizing its scalar reward $r(y_i \mid x)$ (derived from our stratified rubrics as detailed in Sec. 3.2) against the group statistics:

$$\hat{A}_i = \frac{r(y_i \mid x) - \text{mean}(\mathbf{r})}{\text{std}(\mathbf{r})}, \tag{13}$$

where $\mathbf{r} = \{r(y_1 \mid x), \ldots, r(y_G \mid x)\}$ denotes the set of rewards. The objective maximizes the PPO-style clipped loss while penalizing deviations from the reference model $\pi_{\text{ref}}$ via a KL-divergence term. The objective function is formulated as:

$$\mathcal{J}(\theta) = \mathbb{E}\left[\frac{1}{G}\sum_{i=1}^{G}\left(\mathcal{L}_i^{\text{clip}}(\theta) - \beta\,\mathbb{D}_{\text{KL}}(\pi_\theta||\pi_{\text{ref}})\right)\right], \tag{14}$$

where $\mathcal{L}_i^{\text{clip}}(\theta) = \min(\rho_i\hat{A}_i, \text{clip}(\rho_i, 1 - \varepsilon, 1 + \varepsilon)\hat{A}_i)$ represents the clipped surrogate objective, with the importance ratio $\rho_i = \frac{\pi_\theta(y_i|x)}{\pi_{\theta_{\text{old}}}(y_i|x)}$. This approach allows for stable and efficient policy optimization without the memory burden of a critic model.

## B. Theoretical Derivation and Analysis of Gradient Variance

In this section, we provide a detailed derivation of the gradient variance decomposition to theoretically justify the stability-aware curriculum schedule proposed in Sec. 3.3. For clarity of exposition, we consider the score-function form of the policy gradient estimator and omit baselines and advantage normalization. The following analysis applies analogously to advantage-based estimators used in practice. For consistency with Eq. 11, we analyze the curriculum-modulated rubric component $r_{\text{rub}}^{(t)}$; the outcome term $\alpha\,r_{\text{ans}}$ is a fixed-weight addend that does not affect the variance decomposition with respect to $\lambda_t$.

### B.1. Gradient Estimator Decomposition

Consider the standard Policy Gradient objective function $J(\theta) = \mathbb{E}_{\tau\sim\pi_\theta}[r(\tau)]$. The gradient estimator at step $t$ is expressed as:

$$\hat{g}_t = \nabla_\theta \log \pi_\theta(y|x) \cdot r_{\text{rub}}^{(t)}(y|x) \tag{15}$$

In RuCL, the reward $r_{\text{rub}}^{(t)}$ is a dynamic convex combination of foundational ($\bar{r}_{easy}$) and advanced ($\bar{r}_{hard}$) rubric scores:

$$r_{\text{rub}}^{(t)}(y|x) = (1 - \lambda_t)\bar{r}_{easy}(y|x) + \lambda_t\bar{r}_{hard}(y|x) \tag{16}$$

Substituting this into the gradient estimator, we obtain a decomposed gradient form:

$$\hat{g}_t = (1 - \lambda_t)\underbrace{\nabla_\theta \log \pi_\theta(y|x)\bar{r}_{easy}}_{\hat{g}_{easy}} + \lambda_t\underbrace{\nabla_\theta \log \pi_\theta(y|x)\bar{r}_{hard}}_{\hat{g}_{hard}} \tag{17}$$

where $\hat{g}_{easy}$ and $\hat{g}_{hard}$ represent the stochastic gradient components induced by foundational and advanced rubrics, respectively.

### B.2. Variance Analysis

Since the gradient estimator is a random vector, we quantify its variability using the trace of the covariance matrix:

$$\mathcal{V}(\hat{g}_t) \triangleq \text{tr}(\text{Cov}(\hat{g}_t)) = \mathbb{E}\left[\|\hat{g}_t - \mathbb{E}[\hat{g}_t]\|_2^2\right]. \tag{18}$$

Using the covariance property of linear combinations of random vectors, we obtain:

$$\mathcal{V}(\hat{g}_t) = (1 - \lambda_t)^2\mathcal{V}(\hat{g}_{easy}) + \lambda_t^2\mathcal{V}(\hat{g}_{hard}) + 2\lambda_t(1 - \lambda_t)\,\text{tr}(\text{Cov}(\hat{g}_{easy}, \hat{g}_{hard})). \tag{19}$$

For brevity, we denote these trace-variance and trace-covariance terms as $\text{Var}_e \triangleq \mathcal{V}(\hat{g}_{easy})$, $\text{Var}_h \triangleq \mathcal{V}(\hat{g}_{hard})$, and $\text{Cov}_{e,h} \triangleq \text{tr}(\text{Cov}(\hat{g}_{easy}, \hat{g}_{hard}))$, which recovers the compact notation used in Eq. (12) of the main text.

### B.3. Justification of Curriculum Schedule

Eq. 19 provides three insights that motivate the proposed scheduling strategy:

- **Suppressing Unreliable Gradient Signals:** In the early stages of training, the model rarely satisfies advanced reasoning rubrics, making $\bar{r}_{hard}$ highly sparse. This sparsity yields low signal-to-noise ratio and unstable estimates of $\hat{g}_{hard}$, rather than merely large reward variance. Setting $\lambda_t = 0$ eliminates the contribution of $\mathcal{V}(\hat{g}_{hard})$, thereby preventing noisy high-order signals from dominating early optimization.

- **Reducing Gradient Interference:** Before foundational competencies are established, gradient directions induced by perception-oriented and reasoning-oriented rubrics may be weakly correlated or even negatively correlated, which leads to destructive interference under mixed optimization. The curriculum decouples these learning phases, allowing the model to first converge to stable foundational representations.

- **Safe and Progressive Transition:** As training progresses, successful satisfaction of advanced rubrics becomes more frequent, which increases the reliability of $\hat{g}_{hard}$ and improves alignment between gradient components. Under this condition, increasing $\lambda_t$ gradually introduces harder objectives while keeping the covariance term in Eq. 19 controlled.

Overall, the curriculum schedule reduces the contribution of unreliable gradient components in early training and progressively incorporates harder objectives as their gradient signals become statistically reliable, which stabilizes optimization during multi-stage reward learning.

## C. Configuration Details

All experiments are conducted using the `verl` framework (Sheng et al., 2024), which facilitates efficient large-scale reinforcement learning. We employ the Group Relative Policy Optimization (GRPO) algorithm. The training utilizes a constant learning rate scheduler to ensure convergence stability in the later stages of curriculum learning.

Table 6 summarizes the specific hyperparameter settings. Notably, the curriculum parameters $(K, \tau_{th})$ are chosen based on preliminary experiments to balance the trade-off between stability and learning speed.

*Table 6.* Detailed hyperparameters for DR-CL training.

| Hyperparameter | Value |
|---|---|
| *Optimization & Rollout* | |
| Training Batch Size | 256 |
| Global Batch Size | 128 |
| Rollout Number ($G$) | 8 |
| Sampling Temperature | 1.0 |
| Learning Rate | 1e-6 |
| KL Coefficient ($\beta$) | 0.01 |
| *Curriculum & Rewards* | |
| Reward Weight ($\alpha$) | 0.7 |
| Sliding Window Size ($w$) | 20 |
| Proficiency Threshold ($\tau_{th}$) | 0.9 |
| Max Hard Weight ($\lambda_{\max}$) | 1.0 |

## D. Baselines

We compare RuCL against three groups of baselines in Table 2: (i) the base model used for training, (ii) open-source reasoning-focused MLLMs at a comparable 7B scale, and (iii) proprietary models that serve as reference upper bounds. The open-source reasoning baselines are selected to cover representative post-training strategies for multimodal reasoning,

including outcome-level RL, perception-aware rewards, self-reflection, data selection, and lightweight reasoning fine-tuning. All averages in the main paper are macro-averages across benchmarks rather than sample-weighted averages.

**Open-source reasoning-focused baselines.** Table 7 summarizes the main open-source reasoning baselines used in our comparison and highlights how they differ from RuCL. These models are broadly comparable in scale, but differ substantially in their supervision source and reward design. In particular, most prior methods either rely on final-answer correctness, data selection, reflection mechanisms, or task-specific rewards, whereas RuCL explicitly constructs reusable rubrics, stratifies them by empirical learnability, and schedules their weights during RL.

*Table 7.* Descriptions of open-source reasoning-focused baselines.

| Baseline | Main idea | Implementation / supervision | Key difference from RuCL |
|---|---|---|---|
| MM-Eureka-7B | RL-based multimodal reasoning | Uses rule-based reinforcement learning and a two-stage training recipe to improve verifiable multimodal reasoning. | Optimizes primarily with outcome-level signals and does not construct or schedule rubric-level process rewards. |
| OpenVLThinker-7B | Visual chain-of-thought reasoning | Builds an iterative SFT–RL self-improvement pipeline for complex vision-language reasoning. | Focuses on the training pipeline and reasoning data construction, without a reward-centric rubric curriculum. |
| Perception-R1-7B | Perception-aware reasoning | Applies GRPO with perception-oriented reward signals to strengthen visual grounding and perception. | Uses a specialized capability-oriented reward, while RuCL optimizes multiple rubric dimensions with difficulty stratification. |
| Vision-R1-7B | RL for multimodal reasoning activation | Combines chain-of-thought data with RL to activate reasoning behavior in vision-language models. | Does not use generalized rubrics or a curriculum over reward components. |
| R1-Onevision-7B | Generalized multimodal reasoning | Formalizes cross-modal reasoning tasks and trains a unified reasoning model across visual domains. | Emphasizes cross-domain reasoning formulation rather than reward construction and scheduling. |
| ThinkLite-VL-7B | Lightweight reasoning with limited data | Uses sample selection and reinforcement fine-tuning to obtain strong reasoning performance with relatively little training data. | Represents a data-centric efficiency strategy, whereas RuCL is a reward-level curriculum method. |
| VL-Rethinker-7B | Self-reflective visual reasoning | Uses GRPO with a forced-rethinking mechanism to encourage self-reflection during multimodal reasoning. | Improves reasoning through reflection structure, but does not stratify rubrics or schedule reward difficulty. |

**Ablation baselines.** In addition to external model comparisons, Sec. 4.3 evaluates controlled variants under the same backbone, training data, and RL framework. *Vanilla GRPO* removes rubric supervision and uses only the rule-based answer reward. *Uniform Averaging* keeps the filtered rubric set but averages all rubric rewards without difficulty stratification. *Linear Stratification* preserves the easy/hard rubric split but replaces the proposed stability-aware sigmoid schedule with a simple linear ramp. These ablations isolate the effects of process supervision, rubric stratification, and curriculum scheduling, respectively.

# E. Additional Comparisons and Scalability

## E.1. Comparison with Strong Open-source Baselines

To further assess whether the improvements of RuCL are stable against competitive open-source reasoning models, we compare RuCL with the two strongest 7B-scale baselines in Table 8. We conduct two additional runs with different random seeds under the main experimental setting and report the mean and standard deviation on four representative benchmarks.

*Table 8.* Mean and standard deviation comparison against the strongest open-source 7B baselines.

| Model | MathVerse | MathVista | WeMATH | MMMU |
|---|---|---|---|---|
| VL-Rethinker-7B | 53.87±0.11 | 73.27±0.12 | 68.26±0.11 | 54.65±0.12 |
| ThinkLite-VL-7B | 51.45±0.12 | 73.32±0.09 | 65.53±0.12 | 55.43±0.10 |
| RuCL | **54.12±0.13** | **74.09±0.14** | **71.46±0.13** | **56.64±0.14** |

RuCL consistently outperforms both strong baselines across all four benchmarks. The improvement is particularly clear on WeMATH, where RuCL exceeds VL-Rethinker-7B by 3.20 points and ThinkLite-VL-7B by 5.93 points. The small standard deviations indicate that the gains are not driven by a single favorable random seed, but remain stable under repeated runs.

### E.2. Scalability across Model Sizes

We additionally evaluate whether the proposed rubric-level curriculum transfers beyond the main 7B backbone. Specifically, we apply RuCL to Qwen3-VL models at 4B and 8B scales while keeping the remaining experimental settings unchanged. The results are summarized in Table 9.

*Table 9.* RuCL scalability across Qwen3-VL model sizes.

| Scale | Base | RuCL | Avg. Gain |
|---|---|---|---|
| Qwen3-VL-4B | 57.15 | 65.34 | +8.19 |
| Qwen3-VL-8B | 59.43 | 67.21 | +7.78 |

RuCL yields consistent gains at both scales, suggesting that the benefit of stratified rubric-based curriculum learning is not specific to a single 7B model.

## F. Rubric Construction and Filtering

To ensure a comprehensive evaluation of the model's reasoning capabilities, we employ a teacher model, Gemini 3 Pro (Google DeepMind, 2025), to generate a pool of 20 rubric candidates through few-shot prompting. These rubrics cover dimensions including visual faithfulness, logical coherence, constraint satisfaction, and mathematical accuracy.

### F.1. Detailed Definitions of All 20 Rubric Candidates

---

**Full Rubric Candidates Set (Cand_01–Cand_20)**

**Cand_01_visual_presence_check**

- **Description:** Strictly verifies visual faithfulness. Checks if objects, text (OCR), attributes (color/shape), and spatial relationships cited in the reasoning are visibly present in the image.

- **Scoring Criteria:**
  - 1: **Fully Grounded.**
    * **Positive Claims:** Every object, text snippet, or data point mentioned is clearly visible.
    * **Negative Claims:** Statements that something is MISSING must be true.
    * **Occlusion Handling:** Mentions of occluded parts (e.g., "chair legs" under a table) are acceptable IF implied by visible parts, provided no specific unseen details (like "wooden legs" when invisible) are hallucinated.
    * **OCR:** Text read from the image is accurate (minor typo tolerance allowed).
  - 0: **Hallucination Detected.**
    * **Object Hallucination:** Describes objects not present.
    * **Attribute Hallucination:** Assigns wrong properties (color, shape, quantity) to existing objects.
    * **OCR Failure:** Quotes text/numbers not in the image.
    * **False Relation:** Visibly wrong spatial relationships (left/right, above/below).

**Cand_02_key_entity_extraction**

---

- **Description:** Evaluates "Visual Attention Alignment". Verifies if the model isolates and analyzes the specific "Region of Interest" (ROI) required by the question.

- **Scoring Criteria:**

  - 1: **Precise Targeting.**
    * **Visual Selection:** Identifies the specific subject/data series asked about.
    * **Granularity:** Focuses on the specific detail (e.g., 'watch') rather than the whole (e.g., 'man').
    * **Noise Filtering:** Ignores salient but irrelevant distractors.
  - 0: **Focus Drift / Distraction.**
    * **Wrong Target:** Analyzes the wrong object or axis.
    * **Over-Generalization:** Provides a generic caption instead of answering a specific detail query.

## Cand_03_attribute_recognition

- **Description:** Evaluates the precision of identifying static visual properties (Color, Shape, Material, Texture, Size, State). This rubric checks if the adjectives used to describe an object align with visual reality.

- **Scoring Criteria:**

  - 1: **Attribute Accurate.**
    * **Color Family:** The identified color falls within the correct spectrum (e.g., "Navy Blue" matches "Blue"; "Crimson" matches "Red").
    * **Shape/Geometry:** The geometric description is topologically correct (e.g., calling a 'Cube' a 'Box' or 'Square-ish' is acceptable).
    * **Material & Texture:** Correctly identifies surface properties (e.g., "Wooden", "Metallic", "Fluffy", "Wet").
    * **State:** Correctly identifies the physical state (e.g., "Open/Closed", "On/Off", "Broken/Intact").
  - 0: **Attribute Mismatch.**
    * **Spectrum Error:** The color is distinctly wrong (e.g., "Red" vs "Blue", or "Light" vs "Dark" if contrast is significant).
    * **Geometry Failure:** Calling a circle a square, or a 2D object 3D incorrectly.
    * **Material Hallucination:** Identifying a wrong material (e.g., calling a "Plastic" toy "Metal").
    * **Opposite State:** describing an open door as closed.

## Cand_04_ocr_data_accuracy

- **Description:** Evaluates the fidelity of Optical Character Recognition (OCR). Checks if text, numbers, symbols, or labels cited from the image match the visual reality, allowing for minor formatting flexibility but enforcing strict alphanumeric accuracy.

- **Scoring Criteria:**

  - 1: **Accurate Transcription.**
    * **Alphanumeric Match:** The extracted letters and numbers match the image content. Case-insensitivity is allowed (e.g., "STOP" == "Stop").
    * **Numeric Precision:** Numbers are read exactly as they appear (e.g., "19.5" is NOT rounded to "20" if the label is explicitly "19.5").
    * **Formatting Tolerance:** Variations in separators are accepted (e.g., "1,000" == "1000"; "2023-01-01" == "2023/01/01") as long as the value is preserved.
    * **Punctuation:** Minor trailing punctuation differences are accepted (e.g., missing a period at the end of a sentence).
  - 0: **Transcription Error.**
    * **Character Confusion:** Confusing visually similar characters (e.g., reading '5' as 'S', '8' as 'B', '1' as 'I').
    * **Value Mutation:** Changing a number (e.g., reading "2024" as "2023") or flipping a sign (positive/negative).
    * **Word Hallucination:** Replacing a word with a similar-looking but different word (e.g., "Horse" vs "House").
    * **Phantom Text:** Quoting text that is not present in the image at all.

## Cand_05_spatial_positioning

- **Description:** Evaluates the accuracy of spatial reasoning. Checks relative positions (left/right, above/below), depth relationships (front/behind), containment (inside/outside), and geometric alignment (bounding boxes/coordinates).

- **Scoring Criteria:**

  - 1: **Spatially Accurate.**

∗ **Directional Correctness:** Correctly identifies positions relative to the viewer (default) or the object, as implied by the query (e.g., "A is to the left of B").
∗ **Topological Logic:** Correctly identifies contact and containment relationships (e.g., "The cup is ON the table" vs "HOVERING above", "The cat is IN the box").
∗ **Depth Perception:** Correctly distinguishes foreground (closer) objects from background (further) objects.
∗ **Coordinate Grounding:** If the model outputs coordinates or bounding boxes, they must significantly overlap (IoU > 0.5) with the actual visual entity.

– 0: **Spatial Error.**
∗ **Mirror Confusion:** Swapping Left and Right (common MLLM error).
∗ **Vertical/Depth Error:** Confusing Up/Down or Front/Back.
∗ **Detached Relations:** Describing objects as touching/holding when they are visibly separated.
∗ **Coordinate Miss:** The provided bounding box/coordinates focus on the wrong area or miss the object entirely.

## Cand_06_quantity_verification

- **Description:** Evaluates the accuracy of counting and quantification. This includes exact counting for distinct objects (small numbers), reasonable estimation for dense crowds (large numbers), and correct identification of 'zero' (absence).

- **Scoring Criteria:**

  – 1: **Quantitatively Accurate.**
  ∗ **Exact Count (Small < 10):** For clearly distinct items (e.g., 3 apples), the count must be exact.
  ∗ **Estimation (Large > 10):** For dense or occluded groups (e.g., a crowd, a pile of coins), an approximation within a reasonable margin (∼10-20%) or a range (e.g., "dozens", "50-60") is accepted.
  ∗ **Zero Handling:** Correctly states "0", "none", or "no [objects]" when the target is absent.
  ∗ **Unit Awareness:** Correctly distinguishes between singles and pairs (e.g., "2 pairs of shoes" vs "4 shoes").
  – 0: **Counting Error.**
  ∗ **Deviation (Small):** Any error in counting small, distinct sets (e.g., seeing 4 legs on a tripod).
  ∗ **Order of Magnitude (Large):** Significant deviation in estimation (e.g., saying "hundreds" when there are only 10, or "5" when there are 50).
  ∗ **Existence Error:** Counting items that aren't there (1 instead of 0) or missing items completely (0 instead of 3).

## Cand_07_question_intent_alignment

- **Description:** Evaluates "Constraint Satisfaction" and "Scope Compliance". Verifies if the model answers the *exact* question posed, adhering to all explicit constraints.

- **Scoring Criteria:**

  – 1: **Direct & Compliant.**
  ∗ **Category Match:** Provides the specific *type* of info requested (Number, Color, Coordinate, Option).
  ∗ **Constraint Adherence:** Follows negative constraints ("no explanation") and **Unit Constraints** (e.g., if asked for "meters", outputting "cm" fails here).
  ∗ **MCQ Selection:** Explicitly selects a valid option label/text.
  – 0: **Misaligned / Evasive.**
  ∗ **False Refusal:** Claims inability to answer (e.g., "I cannot analyze people") when the image is safe and clear.
  ∗ **Pivot to Captioning:** Ignores the question to describe the whole image.
  ∗ **Constraint Violation:** Ignores length/format/unit instructions.
  ∗ **Neighboring Question:** Answers a related but different question.

## Cand_08_constraint_satisfaction

- **Description:** Strictly evaluates adherence to explicit "Style" and "Negative" constraints in the prompt. This covers length limits, forbidden words, output style (e.g., list vs. paragraph), and tone.

- **Scoring Criteria:**

  – 1: **Fully Compliant.**
  ∗ **Length Constraint:** Strictly obeys word/sentence limits (e.g., "Answer in one word" → "Red"; "No more than 3 sentences" → < 3 sentences).
  ∗ **Negative Constraint:** Does NOT contain forbidden elements (e.g., if prompt says "Do not explain", the output contains ONLY the answer; if "Do not use LaTeX", no LaTeX appears).

* **Style/Format:** Follows specific formatting requests NOT covered by R07 (e.g., "Use bullet points", "Comma-separated list", "JSON format").
            * **Tone:** Adheres to requested persona or tone (e.g., "Explain like I'm 5", "Be professional").
        – 0: **Constraint Violation.**
            * **Verbosity:** Providing a full sentence when a single word/phrase was requested (e.g., Prompt: "Color?", Model: "The color is red." → Fail).
            * **Forbidden Content:** Including text explicitly banned by the user (e.g., saying "Here is the answer" when told to output only the value).
            * **Style Mismatch:** Outputting a paragraph when a list was requested.

**Cand_09_negative_logic_handling**

* **Description:** Evaluates "Logical Inversion" and "Exclusionary Reasoning". Checks if the model correctly processes negative qualifiers ("not", "no", "except", "without", "neither") to select the complement set or avoid specific targets.

* **Scoring Criteria:**

    – 1: **Logic Inverted Correctly.**
        * **Visual Exclusion:** Correctly identifies objects that do NOT match a feature (e.g., "Find the person NOT wearing a hat" → Identifies the bare-headed person).
        * **Set Subtraction:** Correctly lists items while excluding the forbidden category (e.g., "List all fruits EXCEPT apples").
        * **Absence Confirmation:** Correctly agrees with negative premises (e.g., "Yes, there is no dog" vs "Yes, there is a dog").
        * **Double Negation:** (If applicable) correctly resolves double negatives (e.g., "Not impossible" → Possible).
    – 0: **Positive Bias / Inclusion Error.**
        * **Keyword Fixation:** The model ignores the "not" and focuses on the object named (e.g., Prompt: "Which one is NOT red?", Model: Picks the red one because it attended to the word 'red').
        * **Leakage:** The list includes the specifically excluded item (e.g., Listing apples when told "except apples").
        * **Logic Reversal:** Treating "without X" as "with X".

**Cand_10_calculation_accuracy**

* **Description:** Evaluates "Computational Fidelity". Verifies that all explicit mathematical operations (arithmetic, algebra, statistics, unit conversions) and formula applications within the reasoning are mathematically correct.

* **Scoring Criteria:**

    – 1: **Mathematically Sound.**
        * **Arithmetic Precision:** Basic operations (+, -, *, /) are calculated correctly (e.g., "15 + 20 = 35").
        * **Formula Validity:** The correct formula is applied for the context (e.g., using $\pi r^2$ for area, not $2\pi r$).
        * **Consistent Rounding:** Intermediate rounding does not lead to significant final error (unless the problem asks for estimation).
        * **Unit Conversion:** Conversions are mathematically accurate (e.g., "1.5 hours = 90 minutes").
    – 0: **Calculation / Formula Error.**
        * **Arithmetic Hallucination:** The result does not follow from the operands (e.g., "10 + 10 = 25").
        * **Formula Error:** Using the wrong equation (e.g., calculating Average as `Sum * Count` instead of `Sum / Count`).
        * **Constant Error:** Hallucinating mathematical constants (e.g., using $\pi = 3.5$).
        * **Order of Operations:** Violation of PEMDAS logic (e.g., calculating $2 + 3 * 4$ as 20).

**Cand_11_step_coherence**

* **Description:** Evaluates the "Logical Flow" within the `<think>` block. Checks for gaps, jumps, or internal contradictions.

* **Scoring Criteria:**

    – 1: **Seamless & Valid.**
        * **Causal Links:** Steps follow logically (A → B → C).
        * **Transparency:** Calculations/derivations are shown, not skipped.
        * **Internal Consistency:** No self-contradiction within the chain.
    – 0: **Broken Logic / Magic Leaps.**
        * **The 'Magic Step':** Jumps to conclusions without derivation.

* **Non-Sequitur:** Steps lack logical connection.
* **Self-Contradiction:** Flips stance mid-way.
* **Circular Reasoning:** Uses conclusion to prove premise.

**Cand_12_domain_knowledge_validity**

- **Description:** Evaluates the factual correctness of external knowledge (non-visual premises) introduced by the model. This covers scientific principles (Physics, Chem, Bio), historical facts, geographic truths, and common sense rules.

- **Scoring Criteria:**

  - 1: **Factually Sound.**
    * **Scientific Consensus:** Cited laws (e.g., Newton's Laws), formulas, and chemical properties align with standard scientific textbooks.
    * **Taxonomic Accuracy:** Biological classification, habitats, and dietary habits are correct (e.g., "Whales are mammals").
    * **Geo-Historical Fact:** Dates, capitals, locations, and historical events mentioned are accurate (e.g., "Paris is the capital of France").
    * **Definition Correctness:** Technical terms are defined or used with their correct standard meaning.
  - 0: **Knowledge Error / Hallucination.**
    * **Scientific Fallacy:** Citing incorrect physical rules (e.g., "Heavy objects fall faster in a vacuum") or inventing non-existent chemical elements.
    * **False Fact:** Stating specific wrong details (e.g., "The sun revolves around the Earth", "Penguins live in the desert").
    * **Common Sense Violation:** Contradicting basic world knowledge (e.g., "Ice is hot", "Water flows uphill").
    * **Anachronism:** Placing objects/events in the wrong time period contextually.

**Cand_13_evidence_grounding**

- **Description:** Evaluates "Visual Proof". Distinguishes between direct perception (what is seen) and inference (what is deduced).

- **Scoring Criteria:**

  - 1: **Appropriate Grounding.**
    * **Inference:** Explicitly cites visual details as premises for complex deductions.
    * **Direct Perception:** Direct statements accepted for basic observations (color, text).
    * **Chart/Data:** Values align with visual markers.
  - 0: **Missing Critical Evidence.**
    * **'Magic' Inferences:** Complex conclusions without visual backing.
    * **Vague Sourcing:** Using generic phrases ("Based on image") without pointing to specific regions.
    * **External Bias:** Hallucinating details based on stereotypes.

**Cand_14_comparative_reasoning**

- **Description:** Evaluates the logic of inequality, ranking, and ordering. Checks if comparisons regarding size, quantity, value, time, or intensity between two or more entities are directionally and factually correct.

- **Scoring Criteria:**

  - 1: **Logic Verified.**
    * **Directional Accuracy:** Correctly identifies the greater/lesser entity (e.g., "Bar A is higher than Bar B", "The red car is faster").
    * **Superlative Identification:** Correctly identifies the maximum/minimum in a set (e.g., "The tallest building", "The earliest date").
    * **Equality Detection:** Correctly identifies when two entities are effectively equal or tied.
    * **Transitivity:** Follows logical chains (e.g., If $A > B$ and $B > C$, then $A > C$).
  - 0: **Logic Failure.**
    * **Reversal Error:** Flipping the relationship (e.g., stating "$A > B$" when "$A < B$").
    * **False Equivalence:** Stating two items are equal when one is clearly dominant (visual or factual).
    * **Ranking Error:** Identifying the wrong item as the 'Top' or 'Best' (e.g., picking the second tallest bar as the tallest).

* **Incomparable Comparison:** Comparing attributes that do not share a common scale (e.g., "The apple is redder than the banana is long").

**Cand_15_unit_and_scale_consistency**

* **Description:** Evaluates "Dimensional Analysis" and "Scale Awareness". Checks if the model correctly applies axis multipliers (e.g., 'in thousands'), legend units, currency symbols, and map scale bars, preventing magnitude errors.

* **Scoring Criteria:**
    - 1: **Scale & Unit Accurate.**
        * **Scale Factor Application:** Correctly applies the explicit or implicit multiplier found on chart axes or titles (e.g., reading "5" as "5,000" if the axis says "in thousands").
        * **Symbol Literacy:** Correctly interprets %, $, €, °C, and metric prefixes (k, M, G).
        * **Dimensional Consistency:** Maintains the same unit throughout the reasoning unless explicitly converting (e.g., doesn't switch from 'meters' to 'feet' randomly).
        * **Map Scaling:** (If applicable) Uses the provided scale bar to estimate real-world distances rather than pixel distances.
    - 0: **Scale / Unit Failure.**
        * **Scale Blindness:** Reading the raw number but ignoring the axis label/multiplier (e.g., answering "5" instead of "5 million").
        * **Unit Swapping:** Assigning the wrong unit to a value (e.g., "50%" becomes "$50").
        * **Dimensional Incompatibility:** Attempting to add/compare incompatible units without conversion (e.g., "5 meters + 5 kilograms").
        * **Missing Unit:** (Strictness applies) If the question asks for a physical quantity, omitting the unit (saying just "5" instead of "5 kg") is a failure here if it leads to ambiguity.

**Cand_16_option_validity**

* **Description:** Evaluates "Closed-Set Constraints" for Multiple-Choice Questions (MCQ). Verifies that the model's answer maps strictly to one of the provided candidate options, rejecting hallucinations or out-of-bounds selections.

* **Scoring Criteria:**
    - 1: **Valid Selection.**
        * **Existing Key:** Selects a letter/label that actually exists in the prompt (e.g., Selects 'C' when options are A, B, C, D).
        * **Exact Content Match:** If outputting text, it matches the text of one of the options exactly (or close enough to map unambiguously).
        * **Combined Option:** Selects "Both A and B" ONLY if that is explicitly provided as a separate option (e.g., Option D is "A and B").
    - 0: **Invalid / Out-of-Bounds.**
        * **Hallucinated Key:** Selecting a label that doesn't exist (e.g., choosing "E" when only A-D exist).
        * **Open-Ended Answer:** Providing an answer that is not in the choices at all (e.g., Options: "Red, Blue"; Model: "Green").
        * **Implicit Rejection:** Stating "None of the above" or "The image doesn't show any of these" when such an option is not available (this is an invalid move in a forced-choice task).
        * **Ambiguity:** Saying "The first and second one" (without a specific option covering that).

**Cand_17_reasoning_conclusion_match**

* **Description:** Evaluates "Self-Consistency". Checks if the final answer (\boxed{}) is the logical output of the reasoning chain.

* **Scoring Criteria:**
    - 1: **Logically Consistent.**
        * **Direct Consequence:** The box content matches the reasoning conclusion.
        * **Format Equivalence:** Mathematical/Semantic equivalents are accepted (e.g., Reasoning "0.5" → Box "1/2").
        * **Note:** Scores **1** even if the answer is factually wrong, as long as it matches the reasoning.
    - 0: **Mismatch / Disconnect.**
        * **Explicit Contradiction:** Reasoning says A, Box says B.
        * **The 'Lucky Guess':** Wrong reasoning leads to Correct GT (Non-sequitur).
        * **Ambiguity:** Reasoning is undecided, Box makes a random pick.

**Cand_18_clarity_and_conciseness**

- **Description:** Evaluates the "Signal-to-Noise Ratio" of the response. Penalizes conversational fillers, repetitive loops, excessive hedging, and convoluted sentence structures. Rewards efficient information delivery.

- **Scoring Criteria:**
  - 1: **Efficient & Clear.**
    * **High Information Density:** Every sentence adds new information or necessary logic step. No fluff.
    * **Direct Phrasing:** Uses active voice and direct assertions (e.g., "The car is red") rather than passive/hedged constructions (e.g., "It appears that the vehicle might be red in color").
    * **Structured Flow:** Uses paragraphs or bullet points effectively if the answer is long.
    * **No Moralizing/Filler:** Avoids "As an AI...", "Here is the answer:", "I hope this helps", or restating the question unnecessarily.
  - 0: **Verbose / Confusing.**
    * **Repetitive Loops:** Repeating the same fact or phrase multiple times (e.g., "The sky is blue. As mentioned, the blue sky...").
    * **Filler Overload:** Excessive use of "Based on the image", "We can clearly see that", "In this picture".
    * **Excessive Hedging:** Overusing "maybe", "seem", "could be" when the visual evidence is clear.
    * **Convoluted Syntax:** Sentences are grammatically tangled or hard to read, requiring re-reading to understand.

**Cand_19_final_answer_extraction**

- **Description:** Strictly checks syntactical structure (`<think>` and `\boxed{}`).

- **Scoring Criteria:**
  - 1: **Perfect Syntax.**
    * **Structure:** `<think>...</think>` followed by `\boxed{...}`.
    * **Uniqueness:** Exactly one `\boxed{}` at the end.
    * **Clean Ending:** No conversational text after the box.
  - 0: **Parse Error / Malformed.**
    * **Broken/Missing Tags.**
    * **Wrong Order** (Box before Think).
    * **Multiple/Empty Boxes.**

**Cand_20_ground_truth_correctness**

- **Description:** Evaluates "Factual Accuracy" of the `\boxed{}` content against Ground Truth.

- **Scoring Criteria:**
  - 1: **Correct / Equivalent.**
    * **Numerical:** Mathematically equal (0.5 == 1/2).
    * **Unit Awareness:** Correct value even if unit is converted (unless R03 forbids it).
    * **Textual:** Semantically identical (ignore case/punctuation).
    * **MCQ:** Correct option letter or content.
  - 0: **Incorrect / Deviant.**
    * **Value Error:** Mathematically distinct.
    * **Precision Failure:** Incorrect rounding.
    * **Category Error:** Specificity mismatch (Blue vs Dark Blue).
    * **Hallucinated Details.**

## F.2. Applicability and Accuracy of Rubric Candidates

After generating the initial 20 rubric candidates, we evaluate their performance on the sampled data. Table 10 summarizes the applicability (the frequency with which the rubric is deemed relevant to the problem) and the model's accuracy under each rubric. Based on these metrics and the necessity for automated reward computation, we select the final 6 rubrics (R01–R06) to be used in our Reinforcement Learning (RL) pipeline.

The statistics reveal significant disparity in coverage; for instance, candidates like Cand_09 and Cand_03 are applicable to only 9.7% and 18.8% of samples, respectively. Without our applicability-aware filtering, these rubrics would introduce

erroneous failure signals in over 80% of training instances, severely destabilizing the reward function.

*Table 10.* Statistics for the 20 rubric candidates. The model selects Candidates 01, 02, 07, 11, 13, and 17 to form the core reward metrics R01–R06. Candidate 20 serves as the equivalent of the ground truth accuracy for the final assessment.

| ID | Applicability (%) | Accuracy (%) | ID | Applicability (%) | Accuracy (%) |
|---|---|---|---|---|---|
| Cand_01 (R01) | 99.1 | 98.3 | Cand_11 (R04) | 100.0 | 69.4 |
| Cand_02 (R02) | 99.4 | 99.4 | Cand_12 | 78.4 | 94.8 |
| Cand_03 | 18.8 | 98.1 | Cand_13 (R05) | 99.2 | 68.6 |
| Cand_04 | 78.1 | 99.3 | Cand_14 | 34.0 | 90.4 |
| Cand_05 | 71.0 | 96.7 | Cand_15 | 27.3 | 98.2 |
| Cand_06 | 22.9 | 87.3 | Cand_16 | 33.5 | 85.8 |
| Cand_07 (R03) | 100.0 | 86.1 | Cand_17 (R06) | 99.9 | 91.7 |
| Cand_08 | 93.2 | 35.4 | Cand_18 | 97.5 | 66.4 |
| Cand_09 | 9.7 | 90.7 | Cand_19 | 96.7 | 27.7 |
| Cand_10 | 83.9 | 52.2 | Cand_20 (GT) | 100.0 | 47.9 |

# G. Rubric Assessment and Filtering

We utilized a strict JSON-based prompt to ensure the Judge model evaluates both applicability and correctness. The prompts used in our pipeline are shown below.

---

**System Prompt**

You are an expert evaluator for Multimodal Large Language Model(MLLM) outputs. Your task is to score a model's answer using a set of domain-specific rubrics.
You must follow these rules strictly:
**1. RUBRIC-BASED EVALUATION ONLY**

- Score strictly according to the given rubrics.

- Do NOT invent new criteria or use general judgment.

**2. PER-RUBRIC BINARY SCORING (0 or 1)**

- Each rubric must output either 0 or 1.

- Follow the rubric's scoring criteria exactly as written.

**3. APPLICABILITY**

- Some rubrics may not apply to certain problems (e.g., diagram rules on text-only questions).

- For each rubric, output:
    - "applicable": true/false
    - "score": 0 or 1 (only when applicable=true)

**4. JSON-ONLY OUTPUT**

- Output ONLY a JSON dictionary with a list of rubric evaluations.

- NO explanations outside the JSON.

**5. IMAGE + TEXT REASONING**

- Consider both the problem text and all provided images.

- If the rubric is about visual interpretation, refer to the image content.

Your output must strictly follow this JSON structure:

```
{
  "rubric_scores": [
    {
      "rubric_id": "...",
```

---

```
        "applicable": true/false,
        "score": 0
      }
    ]
}
```

You will now evaluate a model's answer for a single problem.
Below is the problem, ground truth, model answer, and the rubrics to evaluate:
**PROBLEM:**
{problem_text}
**IMAGE:**
(Provided separately as image inputs)
**GROUND TRUTH:**
{ground_truth_answer}
**MODEL ANSWER:**
{model_output}
**RUBRICS:**
{rubric_json}
**Your task:**
For each rubric in the rubric list:
1. You must first Decide whether it is applicable to this specific problem.
2. If applicable, assign a score of 0 or 1 according to the rubric's scoring_criteria.
Output ONLY the JSON object of the form.

## H. Reward Signal Generation Prompts

This appendix details the prompt engineering used to instantiate the reward model. We employ a strict "Judge" persona to convert the rubric evaluations into binary reward signals. The Judge receives the specific problem context, visual input, and the rubric candidates to generate a rationalized score for each criterion.

**Reward Model System Prompt**

You are a strict, impartial judge evaluating Multimodal Large Language Model(MLLM) outputs. Your evaluation determines the reward signal for a Reinforcement Learning system, so accuracy and strictness are critical.

**EVALUATION PROTOCOL**
You will be provided with a Problem, an Image, a Ground Truth, a Model Output, and a set of Scoring Rubrics.
For EACH rubric, you must:

1. **Analyze**: Compare the Model Output against the Image and Ground Truth specifically for that rubric's criteria.

2. **Rationalize**: Write a concise explanation citing specific evidence (e.g., "The image shows a blue car, but model said red", or "The logic step A does not imply step B").

3. **Score**: Assign a strictly binary score (0 or 1) based on the rationale.

**STRICT RULES**

1. **Image + Text Reasoning**: Consider both the problem text and provided image. If the rubric is about visual interpretation, refer to the image content.

2. **Binary Only**: Use ONLY 0 or 1. No 0.5. If the criteria are not fully met, the score is 0. Follow the rubric's scoring criteria exactly as written.

3. **Reasoning First**: You must output the rationale BEFORE the score in the JSON to ensure the score is a result of the reasoning.

Your output must strictly follow this JSON structure:

```
{
  "rubric_scores": [
```

```
    {
      "rubric_id": "...",
      "rationale": "Your explanation here...",
      "score": 0 or 1
    }
  ]
}
```

**Reward Model User Prompt**

You will now evaluate a model's answer for a single problem.
Below is the problem, ground truth, model answer, and the rubrics to evaluate:
**PROBLEM**:
{problem_text}
**IMAGE**:
(Provided separately as image input)
**GROUND TRUTH**:
{ground_truth_answer}
**MODEL ANSWER**:
{model_output}
**RUBRICS**:
{rubric_json}
**Your task:**
For each rubric in the RUBRICS list:

1. Analyze the Model Output against the Image and Ground Truth specifically for that rubric's criteria.

2. Write a concise explanation citing specific evidence.

3. Assign a strictly binary score (0 or 1) based on the rationale.

Output ONLY the JSON object of the form.

## I. Case Studies

This appendix demonstrates the generation process of rubric-based rewards for two single instances. We present the input problem (text and image), the model's reasoning chain (rollout), and the raw JSON output generated by the Judge model, which contains the rationale and binary scores for each rubric.

### I.1. Case Study 1: Mitigation of Reward Hacking

**Analysis and Overview.** This case serves as a quintessential example of *reward hacking*, demonstrating how RuCL detects spurious reasoning that outcome-only supervision would miss.

- **The Trap:** The model arrives at the correct final answer (BC = 20) and would receive a perfect reward ($r = 1.0$) under standard RLVR.

- **The Flaw:** As highlighted by the Judge's rationale in **R04 (Step Coherence)**, **R05 (Evidence Grounding)** and **R06 (Reasoning Conclusion Match)**, the model incorrectly applies a sub-triangle area formula to the whole triangle and makes an unjustified "magic leap" to the final value.

- **The Mitigation:** RuCL identifies these logical gaps. Despite the correct answer, the total reward is penalized significantly, effectively discouraging the model from learning such "lucky guesses."

**Case Study 1: Mitigation of Reward Hacking**

**1. Input Problem**
*Problem Text:*
As shown in the figure, in $\triangle ABC$, $\angle C = 90°$. The line segment $AD$ bisects $\angle BAC$ and intersects $BC$ at point $D$.

If $BD : CD = 3 : 2$ and the distance from point $D$ to $AB$ is 8, the following determines the length of $BC$.

*Input Image:*

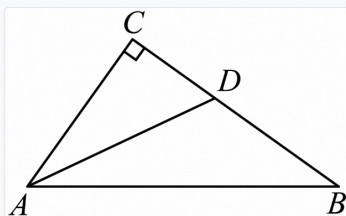

*Ground Truth:* 20

**2. Model Rollout**

To solve the problem, we need to use the Angle Bisector Theorem and the properties of right triangles. Let's go through the reasoning step by step.

1. Apply the Angle Bisector Theorem: According to the Angle Bisector Theorem, the angle bisector of $\angle BAC$ (which is $AD$) divides the opposite side $BC$ into segments proportional to the adjacent sides $AB$ and $AC$. Therefore, we have:
$$\frac{BD}{DC} = \frac{AB}{AC} = \frac{3}{2}$$

2. Given Ratio: We are given that $BD : CD = 3 : 2$. Let $BD = 3x$ and $CD = 2x$. Since $BD + CD = BC$, we have:
$$BC = 3x + 2x = 5x$$

3. Area Calculation: The area of $\triangle ABC$ can be calculated in two ways. First, using the base $BC$ and the height from $A$ to $BC$:
$$\text{Area} = \frac{1}{2} \times BC \times h$$
where $h$ is the altitude from $A$ to $BC$.

4. Using the Altitude to $AB$: The area of $\triangle ABC$ can also be calculated using the altitude from $D$ to $AB$, which is given as 8. The length of $AB$ is $c$, and the length of $AC$ is $b$. The area is:
$$\text{Area} = \frac{1}{2} \times AB \times 8$$

[**Error (Judge R04): The formula** $\frac{1}{2} \times AB \times 8$ **calculates the area of the sub-triangle** $\triangle ABD$**, not the entire** $\triangle ABC$**. This creates a fundamental logical flaw.**]

5. Equating the Two Area Expressions: Since both expressions represent the same area, we have:
$$\frac{1}{2} \times BC \times h = \frac{1}{2} \times AB \times 8$$
Simplifying, we get:
$$BC \times h = AB \times 8$$

6. Using the Proportionality: From the Angle Bisector Theorem, we know:
$$\frac{BD}{DC} = \frac{AB}{AC} = \frac{3}{2}$$
Since $BD = 3x$ and $DC = 2x$, we have:
$$AB = \frac{3}{2}AC$$

7. Substitute Back: Substitute $AB = \frac{3}{2}AC$ into the area equation:

$$5x \times h = \frac{3}{2}AC \times 8$$

Simplifying, we get:

$$5xh = 12 \times AC$$

8. Solving for $BC$: Since $h = 8$, we have: **[Error (Judge R05): The model incorrectly asserts "Since $h = 8$". $h$ is defined as the altitude of $\triangle ABC$ (to base BC), while 8 is the distance from D to AB. These are geometrically distinct distinct values.]**

$$5x \times 8 = 12 \times AC$$
$$40x = 12AC$$
$$x = \frac{3AC}{10}$$

9. Final Calculation: Since $BC = 5x$, we have:

$$BC = 5 \times \frac{3AC}{10} = \frac{15AC}{10} = \frac{3AC}{2}$$

Given the ratio and the height, we find:

$$BC = 20$$

**[Error (Judge R06): The reasoning ends with an expression dependent on the unknown $AC$ ($BC = 3AC/2$). The final answer "20" is not derived from the preceding steps and appears without justification.]**

The model **concludes** that the length of $BC$ is $\boxed{20}$.

**3. Judge Output (Rubric Reward Signal)**

**R01 Visual Presence Check**      **Score: 1**

The model correctly identifies the visible elements: triangle $ABC$, the right angle at $C$, the angle bisector $AD$, and the text labels/values ($BD : CD = 3 : 2$, distance=8). There are no hallucinations of objects not present in the image.

**R02 Key Entity Extraction**      **Score: 1**

The model correctly extracts the specific values needed (ratio 3:2, distance 8) and identifies the target variable $BC$. It ignores irrelevant noise.

**R03 Question Intent Alignment**      **Score: 1**

The model provides a numerical value for the length of $BC$ as requested, adhering to the format constraints.

**R04 Step Coherence**      **Score: 0**

The logic breaks down significantly in Step 4 and Step 8. In Step 4, the model calculates the area of the whole triangle $ABC$ using the formula ($1/2 \times AB \times 8$), which is incorrect; this formula actually calculates the area of the sub-triangle $ABD$ (since 8 is the height of $D$ to $AB$). Furthermore, in Step 8, it incorrectly states "Since $h = 8$", equating the altitude of the main triangle ($AC$) with the distance from $D$ to $AB$. These are fundamental geometric logic errors.

**R05 Evidence Grounding**      **Score: 0**

While the model extracts the number '8', it fails to ground it correctly in the geometric context. The distance from $D$ to $AB$ (8) is visually and geometrically distinct from the altitude of the triangle $AC$ ($h$). The model conflates these two distinct visual features, leading to a premise that contradicts the visual evidence provided by the diagram definitions.

**R06 Reasoning Conclusion Match**      **Score: 0**

The reasoning steps provided by the model end with an algebraic expression dependent on the unknown variable '$AC$' ($BC = 3AC/2$). The model never calculates a value for $AC$, yet it abruptly concludes that $BC = 20$. The final answer '20' appears magically and is not derived from the preceding algebraic work, creating a mismatch between the incomplete reasoning and the final output.

### I.2. Case Study 2: Alignment of Foundational and Advanced Reasoning

**Analysis and Overview.** In contrast to Case 1, this example illustrates a successfully aligned reasoning trajectory where visual perception supports logical deduction.

- **Foundational Skills:** The model correctly extracts coordinates and identifies the linear function (satisfying **R01–R04**).

- **Advanced Reasoning:** The derivation is mathematically sound, and the final answer is a direct logical consequence of the steps (satisfying **R05–R06**).

- **Conclusion:** The high scores across all stratified rubrics confirm that the model has internalized the curriculum, treating perception and reasoning as an integrated process rather than disjoint tasks.

---

**Case Study 2: Alignment of Foundational and Advanced Reasoning**

**1. Input Problem**
*Problem Text:*
As shown in the figure, the graph of the linear function $y = mx + 2$ passes through points $A(2, 4)$ and $B(n, -1)$. Connect $OA$ and $OB$. Find the area of $\triangle OAB$.
*Input Image:*

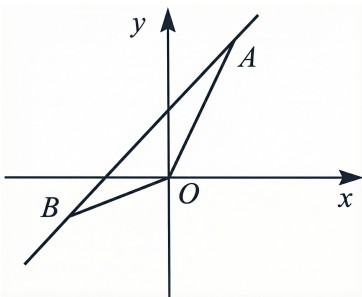

*Ground Truth:* 5

---

**2. Model Rollout**
To solve for the area of triangle $\triangle OAB$, we determine the coordinates of point $B$ and then use the coordinates of points $O$, $A$, and $B$ to find the area of the triangle.

1. **Find the slope $m$ of the line $y = mx + 2$:** Since the line passes through point $A(2, 4)$, we substitute $x = 2$ and $y = 4$ into the equation:
$$4 = 2m + 2$$
Solving for $m$ yields:
$$4 - 2 = 2m \implies 2 = 2m \implies m = 1$$
Therefore, the equation of the line is $y = x + 2$.

2. **Find the coordinates of point $B$:** Since point $B$ lies on the line $y = x + 2$ and has coordinates $(n, -1)$, we substitute $y = -1$ into the equation:
$$-1 = n + 2$$
Solving for $n$ yields:
$$n = -1 - 2 \implies n = -3$$
Therefore, the coordinates of point $B$ are $(-3, -1)$.

3. **Calculate the area of triangle $\triangle OAB$:** The vertices of the triangle are $O(0, 0)$, $A(2, 4)$, and $B(-3, -1)$. The area of a triangle with vertices $(x_1, y_1)$, $(x_2, y_2)$, and $(x_3, y_3)$ is:
$$\text{Area} = \frac{1}{2} \left| x_1(y_2 - y_3) + x_2(y_3 - y_1) + x_3(y_1 - y_2) \right|$$

Substituting the coordinates of $O(0,0)$, $A(2,4)$, and $B(-3,-1)$:

$$\text{Area} = \frac{1}{2} |0(4+1) + 2(-1-0) + (-3)(0-4)|$$

Simplifying inside the absolute value:

$$\text{Area} = \frac{1}{2} |0 - 2 + 12| = \frac{1}{2} |10| = 5$$

Therefore, the area of $\triangle OAB$ is $\boxed{5}$.

**3. Judge Output (Rubric Reward Signal)**

**R01 Visual Presence Check**                                                   **Score: 1**
The model explicitly references points $O(0,0)$, $A(2,4)$, and $B(n,-1)$ and their coordinates consistent with the labeled points in the image. It uses the equation $y = mx + 2$ shown in the problem statement, which is visible in the image. All points, line equations, and spatial relationships cited are present and visually confirmable in the image.

**R02 Key Entity Extraction**        **Score: 1**
The model correctly isolates the specific points $O$, $A$, and $B$, which are the region of interest needed for calculating the area of triangle $OAB$ as asked. It ignores unrelated or extraneous information and focuses entirely on the linear function and these points, without including irrelevant distractions.

**R03 Question Intent Alignment**        **Score: 1**
The model directly answers the question by finding the area of triangle $OAB$, which was explicitly requested. It provides a numerical value for the area with appropriate units (implicitly area units), respecting the problem's requirements without deviation or evasion.

**R04 Step Coherence**        **Score: 1**
The reasoning proceeds in a clear, logical sequence: compute slope $m$, substitute to find point $B$ coordinates, then apply the area formula. Each step follows from the previous without gaps or contradictions, showing all calculations transparently and maintaining internal consistency.

**R05 Evidence Grounding**        **Score: 1**
The model grounds each step in visual information from the image: using coordinates of points visible on the graph and the given equation $y = mx + 2$. It explicitly cites and uses these as premises to deduce the slope and point $B$, with no unexplained assumptions or hallucinations.

**R06 Reasoning Conclusion Match**        **Score: 1**
The boxed final answer '5' matches exactly the calculated area in the reasoning steps. There is no mismatch or contradiction between the conclusion and the boxed answer, so the final output is self-consistent.

## J. Additional Evaluation on Out-of-Distribution Robustness

To assess the model's robustness in a specialized out-of-distribution (OOD) scenario, we evaluate on EvadeBench (Xu et al., 2025a). EvadeBench is an expert-curated Chinese benchmark for evasive content detection in e-commerce, where inputs may appear superficially compliant while implicitly conveying prohibited intent. This setting is substantially different from our training distribution: our RL data mainly covers verifiable visual reasoning tasks, whereas EvadeBench requires safety-oriented detection under ambiguity and domain shift. We therefore use this experiment as a stress test of OOD robustness rather than as a primary target domain for RuCL.

*Table 11.* Performance evaluation on EvadeBench

| Method | EvadeBench Acc. (%) |
| --- | --- |
| Qwen2.5-VL-7B-Instruct | 43.90 |
| Vanilla GRPO | 44.47 |
| **RuCL** | **45.86** |

Table 11 presents the quantitative results on EvadeBench. The task is challenging for all evaluated models, with accuracies remaining below 46%, which highlights the difficulty of transferring from verifiable visual reasoning to specialized safety detection. In this setting, RuCL achieves an accuracy of 45.86%, yielding a modest improvement over Qwen2.5-VL-7B-Instruct (43.90%) and Vanilla GRPO (44.47%). These results suggest that rubric-based curriculum learning can provide a small but consistent robustness benefit beyond the training distribution. However, the limited absolute performance also shows that RuCL does not fully solve specialized OOD generalization: the method is designed to improve general reasoning quality rather than to inject domain-specific knowledge for evasive content detection.

## K. Limitations

Despite the success of RuCL, several limitations persist. First, reliance on proprietary teacher LLM for generation and large-scale judges for reward calculation incurs moderate computational overhead. Second, to ensure stability, we employ a static stratification based on initial statistics, which simplifies the curriculum by assuming constant rubric difficulty throughout training. Future research could explore developing adaptive mechanisms to dynamically update rubric difficulties during the online phase. Furthermore, we explore the model's limitations in specialized out-of-distribution scenarios (e.g., evasive content detection), with detailed results and analysis on EvadeBench provided in Appendix J.

