# OpenReview forum: "RuCL: Stratified Rubric-Based Curriculum Learning for Multimodal Large Language Model Reasoning"
_ICML.cc/2026/Conference — ICML 2026 regular_

### Official Review · Reviewer_h8Hg · 2026-02-26

**Soundness:** 3
**Presentation:** 3
**Significance:** 4
**Originality:** 3
**Overall Recommendation:** 4
**Confidence:** 2

**Summary:**

The paper proposes RuCL, a stratified rubric-based curriculum learning framework for reinforcement fine-tuning of multimodal LLMs. It builds a reusable pool of generalized rubrics, stratifies them into foundational and advanced levels using applicability filtering and pass-rate statistics, and schedules their weights dynamically based on model proficiency measured over a sliding window. Integrated with an answer verifier and trained via GRPO, RuCL reports strong gains over a Qwen2.5-VL-7B baseline across seven visual reasoning benchmarks.

**Compliance With Llm Reviewing Policy:**

Affirmed.

**Final Justification:**

My issues are generally solved and I will maintain my scores.

**Key Questions For Authors:**

1. Can you provide the exact count and examples of the final rubric pool after applying the filter?
2. Can you provide more details on how generalized rubrics are actually proposed?
3. During the generalized rubric construction phase, how many teacher calls are required per dataset to generate a representative pool of candidates?

**Limitations:**

see weakness

**Strengths And Weaknesses:**

### **Soundness**
* **Strengths:**
    * **Technically Sound Method:** The formulation of rubric aggregation by tiers, governed by a stability-aware trigger and a sliding-window proficiency metric, is a well-reasoned approach to managing the complexity of RL for MLLMs.
* **Weaknesses:**
    * **Strict Applicability Filtering:** The threshold $\tau_{app} = 0.99$ is exceptionally high. The paper lacks a detailed analysis of how many rubrics survive this filter and whether it biases the final reward toward overly generic criteria.

### **Presentation**
* **Strengths:**
    * **Logical Narrative:** The paper is well-structured, clearly moving from the motivations (reward hacking in RLVR) to the two-phase pipeline (construction and dynamic curriculum).
    * **Accessible Formulation:** The equations for the stability-aware trigger and the weight scheduler are easy to interpret, and the high-level figures effectively communicate the core concept.
* **Weaknesses:**
  * **Ambiguity in how generalized rubrics are actually proposed:** The description suggests using instance-level context (image, query, ground truth) to propose candidates, yet the claim is that they are general and shared; more procedural detail is needed.
   * *Wrong legend in Figure 3*: The color for 0.7 should be in blue.

#### **Significance**
* **Strengths:**
    * **Pragmatic Efficiency:** By moving from instance-level rubrics to generalized, amortized rubrics, the framework offers a scalable alternative that is likely to be of high practical utility for large-scale MLLM training.
* **Weaknesses:** No weakness in significnace.

#### **Originality**
* **Strengths:**
    * **Novel Conceptual Shift:** Reframing curriculum learning as a shift in reward design rather than data selection is a fresh and creative perspective for the RLVR community.
    * **Creative Combination:** The integration of stability-aware triggers with tiered rubric rewards is a novel combination of existing ideas tailored specifically for the instability of MLLM reasoning.
* **Weaknesses:**
    * **Differing Contemporary Works:** The work would benefit from a clearer distinction between its reward curriculum and progressive reward shaping [1], which share similar high-level goals. This is acceptable because PRS is a contemporary work.

[1] Enhancing Agentic RL with Progressive Reward Shaping and Value-based Sampling Policy Optimization

---

> ### Author Rebuttal · Authors · 2026-03-31
>
> We sincerely thank the reviewer for the positive assessment of our paper and for recognizing the novelty of reframing curriculum learning from data selection to reward design, as well as the practical value of generalized rubrics. Below we clarify the main questions:
> >W1/Q1 : Strict applicability filtering and details about final rubrics.
>
> Thank you for highlighting the need for more explicit analysis of the filtering process and its impact.
>
> (1) **Number of surviving rubrics.** As specified in Sec 4.1, we generate 20 rubric candidates using the teacher model and evaluate them on a sampled subset of 2,000 instances. After applying the high applicability threshold, **6 rubrics are retained**. Their detailed applicability and pass-rate statistics are reported in Appendix D, and we will move these results to the main paper along with representative examples.
>
> (2) **Effect of strict applicability filtering.** We agree that a high threshold favors high-coverage rubrics. However, this is a deliberate design choice to ensure that reward signals are valid for the majority of training instances. Low-applicability rubrics are only relevant to a small subset of samples and would otherwise produce sparse or misleading rewards, introducing high-variance noise into RL optimization.
>
> (3) **Does this bias toward overly generic criteria?** Empirically, the retained rubrics do not collapse to purely generic checks. Instead, they span a spectrum of difficulty and abstraction: from foundational perception-level criteria to higher-level reasoning constraints. This diversity is further reflected in their pass-rate distribution, which we use to stratify rubrics into different difficulty levels. As a result, the final reward remains both broadly applicable and sufficiently discriminative.
>
> We will include additional statistics (e.g., applicability distribution before/after filtering) and examples in the main paper to further clarify this point.
> > W2/Q2: More details on how generalized rubrics are proposed.
>
> We appreciate the request for clarification. We emphasize that our method performs rubric construction at the dataset level, rather than generating rubrics per instance during RL.
> The process consists of two stages (as described in Sec 3.2):
>
> (1) **Offline proposal**: the teacher model is prompted with representative context (task category, image(s), query, and ground-truth answer) to generate a diverse pool of candidate rubrics that capture reusable reasoning primitives;
>
> (2) **Filtering and stratification**: these candidates are evaluated on sampled rollouts using the base model and judge to estimate applicability and pass rate, after which we retain high-coverage rubrics and stratify them by empirical difficulty.
>
> Importantly, instance-level context is used only in the offline proposal stage to induce a representative candidate pool. The final retained rubrics are shared across all instances during RL, which enables amortized cost and scalability. We will add an explicit algorithm box in the main paper to clearly separate these two stages.
> >W3: Wrong legend in Figure 3.
>
> We thank the reviewer for pointing out this oversight and we will correct this in the revised version.
> > W4: Distinction from Progressive Reward Shaping.
>
> We appreciate the suggestion and agree that the relation to PRS should be clarified better[1]. While both works share the broad goal of making reward learning more effective.
> * PRS progressively shapes reward signals over the optimization process, whereas RuCL focuses on constructing a reusable, stratified rubric space and scheduling reward components according to empirical rubric proficiency.
> * Our curriculum unit is not the sample or a handcrafted shaping stage, but the rubric itself, with stratification driven by applicability filtering and pass-rate-based difficulty estimation.
>
> We will add a dedicated discussion in Related Work to make difference explicit and avoid confusion.
>
> [1] Enhancing Agentic RL with Progressive Reward Shaping and Value-based Sampling Policy Optimization
> >Q3: Number of teacher calls required.
>
> Thank you for raising this question. We clarify that our setting involves a single training dataset (ViRL-39K[1]), which contains diverse categories but does not require separate rubric generation per category.
> * The teacher model is invoked **only once** to construct a global pool of generalized rubric candidates. These rubrics are designed to be **atomic, diverse, and broadly applicable** across different categories, rather than tailored to individual instances or subsets.
> * The number of teacher calls is constant and does not scale with the dataset size or the number of categories.
>
> This design is consistent with our goal of replacing repeated instance-specific rubric generation with a one-time, reusable rubric construction process.
>
> [1] VL-Rethinker: Incentivizing Self-Reflection of Vision-Language Models with Reinforcement Learning

---

> > ### Author Rebuttal · Reviewer_h8Hg · 2026-04-02
> >
> > Thanks for your detailed reply. I will maintain my scores.

---

> > > ### Author Response · Authors · 2026-04-07
> > >
> > > We sincerely thank the reviewer for the careful evaluation and constructive feedback. We appreciate the recognition of the novelty and practical value of our reward-centric curriculum design. The reviewer’s comments have helped us improve the clarity and completeness of the paper, including clarifying the rubric construction process, providing additional statistics and examples for filtering, correcting presentation issues, and strengthening the discussion of related work. We are glad that the concerns have been adequately addressed, and we will further refine the manuscript to enhance its clarity and impact in the revised version.

---

### Official Review · Reviewer_j5AW · 2026-03-11

**Soundness:** 3
**Presentation:** 3
**Significance:** 3
**Originality:** 3
**Overall Recommendation:** 4
**Confidence:** 3

**Summary:**

This paper proposes RuCL, a stratified rubric-based curriculum learning framework for enhancing reasoning in multimodal large language models. It constructs generalized rubrics and stratifies them into foundational and advanced levels based on model competence. It dynamically adjusts rubric weights during training to guide the model from basic visual perception to complex logical reasoning. The method combines rubric-based process rewards with final answer rewards and uses GRPO for optimization. Experiments on seven visual reasoning benchmarks show consistent and significant gains over baseline models, reaching state-of-the-art performance among 7B-scale models.

**Compliance With Llm Reviewing Policy:**

Affirmed.

**Final Justification:**

Although the authors have provided supplementary experiments and explanations, the core issues remain unresolved, so I decide not to raise the score for this paper.

**Key Questions For Authors:**

1.How would you design a dynamic rubric updating mechanism to adjust difficulty stratification during training？
2.How sensitive is the framework to different judge models and rubric sets？

**Limitations:**

yes

**Strengths And Weaknesses:**

Strengths
1.Novel and well-motivated paradigm. It innovatively applies curriculum learning to reward design instead of data selection, effectively solving training instability and reward hacking issues.
2.Solid ablation studies. It validates the effectiveness of stratified rubrics, dynamic scheduling, and reward balancing components clearly.
3.Detailed qualitative analysis. It provides concrete case studies to show how RuCL suppresses spurious reasoning.

Weaknesses
1.Rubric stratification is static. It uses initial performance to split rubric levels and does not adjust difficulty during training.
2.Limited model scalability. Experiments are only conducted on 7B-scale models without testing larger architectures.
3.Judge dependency. The framework relies on a strong judge model, which may limit reproducibility with different judges.
4.Out-of-distribution robustness is limited. Performance on specialized OOD tasks improves only slightly.

---

> ### Author Rebuttal · Authors · 2026-03-31
>
> We thank the reviewer for the positive assessment of the novelty, ablations, and qualitative analysis. We address these concerns as follows:
> >W1/Q1: Rubric stratification is static and design a dynamic updating mechanism.
>
> We thank the reviewer for this insightful suggestion. We agree that a fully static rubric stratification has limitations: as the policy improves during RL, the effective difficulty of a rubric can change, and an online updating mechanism can in principle track the evolving competence boundary more accurately.
>
> * Our main contribution, however, is not a dynamic re-definition of rubric tiers, but a **reward-centric curriculum mechanism** that dynamically adjusts the optimization focus through rubric weighting. In RuCL, the curriculum is already dynamic at the weighting level, while the rubric partition is kept fixed after initialization. We also acknowledge this limitation in the manuscript's discussion of limitations, where we highlight online rubric updating as an important direction for future work.
>
> * We agree that a **dynamic rubric updating mechanism is a natural and promising extension**. A natural extension is an online re-stratification mechanism: every *M* training steps, we recompute each rubric’s EMA-smoothed pass rate on recent rollouts, re-estimate its difficulty, and update its tier only when the change exceeds a hysteresis margin. This would preserve the stability of the current scheduler while allowing the curriculum to adapt to evolving model competence.
>
> We will clarify this limitation in the revision and include dynamic rubric updating as an important future direction.
> >W2: Limited model scalability.
>
> We agree that validating RuCL beyond a single 7B model is important for assessing its scalability and generalizability. Our original experiments focus on the 7B setting because it is the most common open-source benchmark scale for multimodal reasoning RL[1,2].
>
> * To further address this concern, we additionally run RuCL on Qwen3-VL-4B and 8B. Although we are currently unable to extend the rebuttal experiments to the 32B+ regime due to the limited time and computational resources available, the results still provide an important test of whether RuCL transfers across different model sizes and backbone variants, rather than being specific to a single 7B architecture.
>
> * All other experimental settings remain unchanged. As shown in the table below, RuCL consistently yields strong performance gains across both two models, indicating that its effectiveness generalizes across different model scales.
>
> Scale|Base|RuCL|Avg. Gain
> -|-|-|-|
> 4B|57.15|65.34|+8.19
> 8B|59.43|67.21|+7.78
>
> [1] Mm-eureka: Exploring the frontiers of multimodal reasoning with rule-based reinforcement learning
>
> [2] Openvlthinker: Complex vision-language reasoning via iterative sft-rl cycles
> >W3/Q2: Judge dependency and rubric sets sensitivity.
>
> We agree that judge robustness is an important practical issue. RuCL mitigates dependence on any single judge through three design choices: generalized rubrics, applicability-aware evaluation, and a hybrid reward with a deterministic verifier.
>
> * To directly evaluate judge sensitivity, we conduct a controlled experiment where we replace the original judge with **Qwen2.5-VL-72B** while keeping all other settings unchanged. The resulting model achieves a **+7.44** improvement over the base model, compared to **+7.83** in the original setup—a marginal difference of only **0.39**. This close performance suggests that RuCL is robust to the choice of strong judge models and does not critically depend on a specific judge.
>
> * For rubric set sensitivity, we compare the full rubric set (+3.28), Uniform Averaging with filtered rubrics (+4.33, Sec. 4.3), and RuCL (+7.83). As shown in table, unfiltered rubrics introduce noise, filtering helps, and the best results come from filtered rubrics with stratified curriculum rather than any specific rubric set.
>
> |Method|Design|Avg. Gain|
> |-|-|-|
> |Full Rubrics|No filtering / no stratification|+3.28|
> |Uniform Averaging|Filtered, no curriculum|+4.33|
> |RuCL (Ours)|Filtered + stratified curriculum|+7.83|
> >W4: Out-of-Distribution (OOD) Robustness.
>
> We agree that the gain on this specialized OOD task is modest, and we will clarify this point in the revision. EvadeBench is particularly challenging because it requires identifying evasive content that appears superficially compliant but implicitly conveys prohibited intent, which involves strong ambiguity and distribution shift.
>
> This setting is also substantially outside our training distribution. Our RL training data mainly covers verifiable visual reasoning tasks, while EvadeBench targets specialized safety-oriented detection. In addition, RuCL is designed to improve general reasoning quality rather than domain-specific knowledge for evasive content detection. As a result, RuCL can provide a consistent robustness benefit, but it does not fully solve this type of specialized OOD generalization.

---

> > ### Author Rebuttal · Reviewer_j5AW · 2026-04-02
> >
> > The authors provided detailed explanations in their rebuttal and supplemented experiments across different model scales (4B/8B) and judge models (Qwen2.5-VL-72B). However, since the newly added scales are still within the same order of magnitude and the comparative analysis of judge models remains limited, the evidence is insufficient to fully demonstrate the robustness of the method. Regarding the dynamic updating of hierarchical criteria, although the authors clarified that their contribution lies in "weight allocation" rather than "definition reconstruction," I believe critical concerns remain: if the pre-defined "basic" and "advanced" criteria undergo a difficulty inversion during training, would the existing static layering conversely limit the efficiency of curriculum learning?Furthermore, the authors mentioned that RuCL primarily targets verifiable reasoning. However, in multimodal environments, the correctness of the reasoning process is harder to verify and more susceptible to distribution shifts than the final answer. Therefore, I hope the authors can further discuss the following:
> > 1.  Have you observed any performance degradation on simple tasks in the late training stages due to the weights of basic criteria being too low?
> > 2.  In OOD scenarios lacking a deterministic verifier, how does the process reward of RuCL prevent the model from engaging in reward-hacking behavior?

---

> > > ### Author Response · Authors · 2026-04-05
> > >
> > > We thank the reviewer for the thoughtful follow-up questions. The concerns are addressed point by point below.
> > > > Q1:Does performance degrade on simple tasks in the late training stages?
> > >
> > > We agree that this is an important question regarding whether the curriculum remains effective after the optimization focus shifts away from foundational rubrics. As shown in Fig. 3, foundational rubric scores increase rapidly early on and remain stable after reaching a stable level of performance, even as the curriculum shifts focus toward advanced rubrics whose scores continue to improve. This behavior is consistent with our design: the curriculum is performance-triggered and softly weighted, so foundational signals are not removed but only relatively down-weighted after stable mastery. Moreover, advanced rubrics inherently depend on foundational skills, which further prevents degradation. Overall, we do not observe evidence of late-stage collapse on these simple tasks, indicating that the proposed curriculum remains effective throughout training.
> > > > Q2:In OOD settings without a deterministic verifier, how does RuCL’s process reward prevent reward hacking?
> > >
> > > We thank the reviewer for this important question. The reviewer points out a fundamental vulnerability in process supervision: without a deterministic verifier to act as a definitive anchor, the policy model is highly prone to reward hacking. We agree that without a deterministic verifier, process-based rewards can be vulnerable to reward hacking. However, RuCL mitigates this risk through structured design.
> > >
> > > - **Restricting the Optimization Space via Multi-Dimensional Constraints:** Reward hacking typically occurs when a policy exploits a sparse or singular reward signal by finding a spurious shortcut. RuCL addresses this by decomposing the reasoning process into multi-dimensional constraints (e.g., visual perception, evidence grounding). Even without an outcome reward, the policy must simultaneously satisfy these rigorous intermediate checks. Fabricating a reasoning chain that is cohesively structured, visually grounded, and logically sound is much harder for the model than merely guessing a final token or generating ungrounded text.
> > >
> > > - **Aligning Process Rewards via Applicability-Aware Evaluation:** In RuCL, the judge first determines whether a rubric is applicable to the current instance, and only then evaluates whether it is satisfied. This design avoids rewarding irrelevant stylistic patterns or generic “good-looking” reasoning that happens to score well across many examples. Instead, the reward is tied to criteria that are actually required by the current data. This makes the supervision more closely aligned with the reasoning demands of the instance and reduces opportunities for reward hacking.
> > >
> > > - **Reducing Early Overfitting via Curriculum Weighting:** Under OOD shift, higher-level reasoning rubrics are generally more ambiguous and harder than foundational ones. RuCL therefore emphasizes more reliable foundational rubrics in the early stages and only gradually increases the weight on advanced reasoning rubrics after stable competence is reached. This staged optimization reduces overfitting to noisy high-level process signals in the early stages of training.
> > >
> > > Overall, these mechanisms optimize both the judge’s evaluation and the policy’s optimization space, making reward hacking substantially more difficult than in standard process supervision.
> > >
> > > >Q3:Sensitivity to the choice of judge model.
> > >
> > > To address the concern about judge robustness, we provide additional analysis to clarify the impact of the judge model. We first observe a clear performance gap between the two judge models. As shown below, Qwen3-VL-235B consistently outperforms Qwen2.5-VL-72B across multiple benchmarks, indicating that larger judges can provide higher-quality supervision signals.
> > >
> > > Judge Scale|MathVerse|MathVision|MathVista|MMMU
> > > -|-|-|-|-
> > > 72B|59.4|37.1|74.2|71.0
> > > 235B|72.5|59.0|84.9|78.7
> > >
> > > As noted in our previous response, we evaluate the sensitivity of RuCL to the choice of judge by replacing the judge with Qwen2.5-VL-72B while keeping all other settings fixed. As shown below, both RuCL settings achieve consistent improvements over the base model, while the performance gap between them remains small.
> > >
> > > Model|MathVerse|MathVision|MathVista|MMMU
> > > -|-|-|-|-
> > > Base|48.98|24.18|70.20|51.00
> > > RuCL(72B)|53.64|28.60|73.69|56.33
> > > RuCL(235B)|54.14|28.88|74.10|56.67
> > >
> > > These results suggest that although stronger judges provide better supervision signals, the overall training performance is relatively insensitive to the specific choice of judge. This supports the conclusion that the gains of RuCL primarily stem from its rubric-based curriculum design rather than reliance on a particular judge model. This further highlights the robustness and general effectiveness of RuCL across different supervision conditions.

---

### Official Review · Reviewer_qPM3 · 2026-03-12

**Soundness:** 2
**Presentation:** 3
**Significance:** 2
**Originality:** 2
**Overall Recommendation:** 4
**Confidence:** 4

**Summary:**

The paper investigates the use of structured rubrics over model reasoning traces to introduce supplementary rewards that may help improve reasoning and reward learning.

**Compliance With Llm Reviewing Policy:**

Affirmed.

**Final Justification:**

I thank the authors for their effort during the rebuttal. The breadth of the additional experiments they conducted is genuinely commendable.

That said, several aspects remain unclear to me. In particular, I am still not fully convinced about the technical source of the gains. The inference-only pipeline does not appear to yield significant improvements on its own, which suggests that the observed benefits may come specifically from the combination of rubrics and rewards. However, it is still not clear to me why this combination is effective, nor why self-distillation performs worse than the authors’ method.

I also found the sensitivity to the choice of alpha noteworthy. In addition, the figure legends appear to use colors that do not match the corresponding plot lines, which makes the results harder to interpret. More broadly, if careful selection of alpha is important, then it raises the question of whether comparable hyperparameter tuning could also improve competing methods.

Despite these concerns, I am increasing my score to 4, primarily because the empirical results do suggest that the method is effective in practice.

At the same time, I strongly encourage the authors to investigate these issues more carefully. I also believe the paper would benefit from a substantial rewrite to address the many writing and presentation issues raised during the discussion.

**Key Questions For Authors:**

**Questions:**

1.	Could the authors clarify the decoding strategy used for these models? Additionally, reporting standard deviations would help readers assess whether the results are statistically significant, since the baselines currently appear quite close to the reported model performance.

2.	Could the authors analyze whether the judge LLM already outperforms the LLM being optimized? If so, it would be helpful to clarify whether the setting is better understood as directly using the judge model or as a form of distillation.

3.	Could the authors clarify how $\omega_t$ is defined in Equation 2? I may have missed this, but I did not find a clear description of how it evolves over time.

4.	In Equation 9, how is $\alpha$ chosen? Although the optimal value appears to be 0.7, it is unclear what criterion is used to determine this optimum.

5.	In line 297, the green deltas feel somewhat misleading. Deltas are usually reported with respect to the strongest baseline, so it would be helpful to revise or clarify this presentation.

6.	Several recent papers on multi-turn RL appear relevant. Could the authors compare their method to these works ([A, B, C])? The relationship between these directions seems potentially interesting.

7.	Could the authors provide a clearer description of the baselines?

**References:**

[A] Gehring, Jonas, et al. "Rlef: Grounding code llms in execution feedback with reinforcement learning." arXiv preprint arXiv:2410.02089 (2024).

[B] Jain, Arnav Kumar, et al. "Multi-turn code generation through single-step rewards." arXiv preprint arXiv:2502.20380 (2025).

[C] Ekbote, Chanakya, et al. "MURPHY: Multi-Turn GRPO for Self Correcting Code Generation." arXiv preprint arXiv:2511.07833 (2025).

**Limitations:**

Yes, the authors discuss the limitations.

**Strengths And Weaknesses:**

**Strengths:**

1. The paper tackles an interesting problem: obtaining accurate credit assignment for reasoning.

**Weaknesses:**

1.	The results of the paper do not appear to be statistically significant. Since the paper does not report standard deviations, it is difficult to assess this more confidently.

2.	To the best of my knowledge, the paper has a couple of writing issues. In particular, the baselines are not well explained, and some of the mathematical terminology seems underexplained as well (please see question).

3.	The paper appears to implicitly assume that the judge LLM is sufficiently capable of producing a good rubric. However, it does not discuss how the judge LLM itself is evaluated. Moreover, the paper does not investigate whether the judge LLM can already generate the correct answers; if that is the case, it would be useful to clarify whether the setup is essentially a form of distillation.

---

> ### Author Rebuttal · Authors · 2026-03-31
>
> We thank the reviewer for the thoughtful comments and suggestions. The concerns are addressed point by point below:
> >W1/Q1 Decoding strategy and statistical significance.
>
> We agree that the original description about decoding strategy is brief, and we have reported several key parameters in Appendix C. In our implementation, RL rollouts use stochastic sampling for exploration, while evaluation uses deterministic decoding.
>
> To address the concern about statistical significance, we run two additional trials with different random seeds for all ablation methods and report mean ± std over three runs for the average score on the General benchmarks. As shown below, RuCL consistently outperforms other baselines with comparable variance, indicating that the improvements are stable.
> Method|General Avg. (Mean ± Std)
> -|-
> Vanilla GRPO|59.11 ± 0.12
> Uniform Averaging|59.28 ± 0.13
> Linear Stratification|60.81 ± 0.09
> **RuCL**|**63.92 ± 0.11**
>
> We will state this decoding strategy explicitly and include a complete description and statistical analyses in the revised manuscript.
> >W3/Q2: Judge LLM vs. Distillation.
>
> We thank the reviewer for raising this point. We agree that the role and capability of the judge model should be clarified more carefully.
>
> 1. We note that the judge is indeed stronger than the 7B policy model in standalone benchmark performance. We evaluate two judges, Qwen3-VL-235B-A22B and Qwen2.5-VL-72B, on four representative benchmarks:
>
> Scale|MathVerse|MathVision|MathVista|MMMU
> -|-|-|-|-
> 72B|59.4|37.1|74.2|71.0
> 235B|72.5|59.0|84.9|78.7
>
> While the judge serves as a stronger evaluator, we emphasize that RuCL is **not a form of distillation**. The judge does not provide target answer sequences, rationales, or teacher trajectories for imitation. Instead, it only scores sampled policy outputs with rubric-level binary signals, and the policy is updated via GRPO using scalar rewards. Thus, RuCL is more accurately characterized as rubric-based **RL with AI feedback**.
>
> 2. Directly deploying the judge model as the policy corresponds to a fundamentally different setting: it changes inference-time cost and model size, whereas our goal is to improve a much smaller policy through training-time supervision only. To test whether RuCL critically depends on an extremely large judge, we replace the judge with Qwen2.5-VL-72B and rerun training. The final model still achieves a +7.44 average gain over the base model. This suggests that the gain of RuCL is not overly sensitive to the parameter scale of the judge model. The judge mainly serves as a provider of structured rubric-level supervision.
>
> >W2/Q3: Eq. 2 clarifications.
>
> Eq. 2 provides a general multi-objective formulation. In our implementation, the weights $\omega\_k^{(t)}$ are uniform within each group and transition over time solely through the curriculum coefficient $\lambda\_t$.
>
> $\omega_{k}^{(t)} = \begin{cases} \frac{1-\lambda_t}{|\mathcal{R}\_{easy}|}, & R\_k \in \mathcal{R}\_{easy} \\\\ \frac{\lambda_t}{|\mathcal{R}\_{hard}|}, & R\_k \in \mathcal{R}\_{hard} \end{cases}$
>
> Eq. 2  then reduces exactly to Eq. 6 , where the rubric reward is the convex combination of the average scores over the easy and hard strata. We will revise the manuscript to make this explicit.
> >W2/Q4: Parameter $\alpha$.
>
> We treat $\alpha$ in Eq. 9 as the trade-off coefficient between **outcome correctness and rubric-based process supervision**, and choose it by sensitivity analysis over $\alpha \in [0.5, 0.9]$ using the average score across all seven benchmarks. We find that $\alpha = 0.7$ yields the best overall performance.
> >Q5: Green deltas in Table 2.
>
> We appreciate this suggestion. The green deltas are intended to show improvement over the base model. We will relabel this row more explicitly.
> >Q6: Multi-turn RL references.
>
> Thank you for pointing us to these relevant papers. These works and RuCL study different but complementary axes of RL design. RLEF and MURPHY mainly focus on multi-turn optimization with execution feedback / self-correction, while 𝜇Code shows that in some settings multi-turn improvement can even be learned with single-step rewards. By contrast, RuCL focuses on the structure of the reward signal itself: we construct reusable rubrics, stratify them by empirical difficulty, and schedule them through a reward-level curriculum. We will discuss them more explicitly in Related Work.
> >Q7: Baselines explanation.
>
> In the main results, we include three baseline groups: general-purpose open-source MLLMs, reasoning-focused open-source MLLMs, and proprietary models as reference upper bounds. In the ablations, Vanilla GRPO removes rubric supervision, Uniform Averaging removes difficulty stratification, and Linear Stratification replaces our stability-aware schedule with a linear ramp. Together, these baselines isolate the effects of process supervision, rubric stratification, and curriculum scheduling. We will clarify this more explicitly in the revision.

---

> > ### Author Rebuttal · Reviewer_qPM3 · 2026-04-01
> >
> > I thank the authors for the replies. I have a couple of follow up questions:
> >
> > **Q1.** Thank you for the clarification. I wonder whether these additional results should be reported relative to the strongest baselines in the paper, rather than only relative to the base model. In particular, VL-Rethinker-7B appears to be the strongest open-source 7B baseline and therefore seems like the most meaningful comparison. In addition, could the authors clarify how the reported average scores are computed? Are they macro-averaged across benchmarks/datasets, or averaged over all evaluation samples?
> >
> > **Q2.** It is somewhat surprising that the smaller policy achieves performance close to that obtained with a much larger judge. Could the authors clarify if there is any intuition for this? In addition, because the method relies on a stronger model to supervise a smaller model from the same family, it seems important to compare against self-distillation or on-policy distillation baselines. Would such comparisons be more appropriate? Finally, to what extent do the gains come from rubric-based scoring itself, as opposed to the RL update? Are there prompt-optimization or inference-time approaches that could obtain similar benefits under the same rubric feedback?
> >
> > **Q3.** Do the authors observe instances whose difficulty effectively changes during training, for example from easy to hard or from hard to easy? If so, how frequently does this occur, and does it affect the curriculum design?
> >
> > **Q4.** How is $\alpha$ chosen in practice? Is it tuned using a validation set or another model-selection procedure? It would also be helpful to understand how sensitive the reported results are to this parameter.
> >
> > **Q7.** The paper would benefit from a clearer description of the baselines. For each baseline, could the authors explain what it does, how it is implemented, and in what key respect it differs from the proposed method?

---

> > > ### Author Response · Authors · 2026-04-05
> > >
> > > We thank the reviewer for the thoughtful follow-up questions.
> > > >Q1: Comparison to baselines.
> > >
> > > To better demonstrate the performance advantage of RuCL, we compare it against the two strongest baselines, VL-Rethinker and ThinkLite-VL. Under the main experimental settings, we further conduct two additional runs with different random seeds and report the mean and standard deviation across four benchmarks. We clarify that all "Avg." values reported in the paper denote the macro-average across benchmarks.
> > >
> > > As shown below, RuCL consistently outperforms these strong baselines, achieving a significant **+3.20** improvement over VL-Rethinker on WeMATH.
> > > Model|MathVerse|MathVista|WeMATH|MMMU
> > > -|-|-|-|-
> > > VL-Rethinker|53.87±0.11|73.27±0.12|68.26±0.11|54.65±0.12
> > > ThinkLite-VL|51.45±0.12|73.32±0.09|65.53±0.12|55.43±0.10
> > > RuCL|54.12±0.13|74.09±0.14|71.46±0.13|56.64±0.14
> > > >Q2: Self-distillation comparison and RuCL contributions.
> > >
> > > We attribute the significant improvement of the base model to rubric-based curriculum RL and the richer reward signals provided by the judge. Unlike standard knowledge distillation, the judge in RuCL does not provide imitation or distribution-level supervision, but instead progressively provides structured rubric-based rewards. As shown in Sec. 3, these rubric signals effectively act as multi-objective supervision rather than imitation targets.
> > >
> > > A comparison with self-distillation method is also important. We include an additional baseline using an open-source self-distillation approach, SDPO [1], under the same backbone, training data, and evaluation as our method. SDPO relies on distribution matching from a teacher policy, whereas RuCL provides more fine-grained rewards over intermediate reasoning steps. As shown below, RuCL consistently outperforms SDPO across four benchmarks, with the largest gain of **+4.91** on WeMATH.
> > > Method|MathVerse|MathVista|WeMATH|MMMU
> > > -|-|-|-|-
> > > SDPO|52.87|71.82|66.58|54.10
> > > RuCL|54.14|74.10|71.49|56.67
> > >
> > > An additional analysis evaluates whether inference-time rubric feedback alone can yield similar gains. We score the base model’s trajectories across all 7 benchmarks using the same judge and rubrics, and feed the feedback at inference time. This yields only a modest **+2.36** average improvement, substantially smaller than RuCL. Combined with Sec. 4.3, where uniform rubric aggregation improves over vanilla RL but remains weaker than RuCL, this suggests that the gains come from the combination of rubric-based supervision and RL optimization, rather than rubric feedback alone.
> > >
> > > [1] Reinforcement Learning via Self-Distillation
> > > >Q3: Does instance difficulty change during training?
> > >
> > > We agree that the difficulty of some samples may evolve during training as the model improves. However, RuCL focuses on rubric-level stratification rather than instance-level difficulty. As shown in Fig. 3 (left), the scores of foundational rubrics rise rapidly in the early stage and remain stable afterward, even as the scheduler shifts focus toward advanced rubrics. This suggests that the curriculum operates on stable rubric-level signals rather than instance-level difficulty. Importantly, these changes in sample difficulty do not disrupt training. Our curriculum is performance-driven with a soft weighting scheme, reflecting the intended learning process where previously hard behaviors become learnable.
> > > >Q4: Choice and sensitivity of $\alpha$
> > >
> > > In practice, we choose it via a limited sensitivity analysis over $\alpha\in[0.5,0.9]$. We do not introduce an additional model-selection stage. As shown in Sec. 4.3 and Fig. 3 (right), the method is moderately sensitive to $\alpha$: performance improves from lower values, peaks at $\alpha = 0.7$, and then declines when $\alpha$ becomes too large. Empirically, smaller $\alpha$ over-emphasizes rubric-based process supervision, while larger $\alpha$ makes the training closer to sparse outcome-only supervision.
> > > >Q7: Baseline descriptions.
> > >
> > > We provide a description of the open-source reasoning baselines.
> > > Baseline|Method|Implementation|Features
> > > -|-|-|-
> > > MM-Eureka|RL-based multimodal reasoning|rule-based RL+two-stage training|outcome-level RL
> > > OpenVLThinker|visual CoT reasoning|SFT+RL self-improvement|pipeline design, no reward curriculum
> > > Perception-R1|perception-aware reasoning|GRPO+perception reward|single-capability reward
> > > Vision-R1|activates multimodal reasoning|CoT+RL|no rubric-based reward
> > > R1-Onevision|generalized multimodal reasoning|cross-modal formalization|not reward-centric
> > > ThinkLite-VL|reasoning with little data|sample selection+RFT|data-level curriculum
> > > VL-Rethinker|self-reflective reasoning|GRPO+Forced Rethinking|reflection mechanism, no rubric stratification
> > >
> > > Due to space limitations, we will include a more detailed description of the baselines and additional experimental results in the revised version.
> > >
> > > We sincerely thank the reviewer again for the careful reading and valuable suggestions.

---

### Official Review · Reviewer_CeUu · 2026-03-13

**Soundness:** 3
**Presentation:** 3
**Significance:** 3
**Originality:** 2
**Overall Recommendation:** 4
**Confidence:** 3

**Summary:**

This paper proposes RuCL, a rubric-based reinforcement learning framework for improving multimodal reasoning. Instead of rewarding only final-answer correctness, it builds a set of generalized rubrics, splits them into foundational and advanced categories by empirical difficulty, and gradually shifts training emphasis from easier to harder rubric rewards through a dynamic curriculum. Experiments on seven visual reasoning benchmarks show consistent gains over the Qwen2.5-VL-7B baseline and competitive performance against strong open-source reasoning models.

**Compliance With Llm Reviewing Policy:**

Affirmed.

**Final Justification:**

The author has addressed my concern, I will keep my original score.

**Key Questions For Authors:**

Please see weakness

**Strengths And Weaknesses:**

strengths

1 It applies curriculum learning to the reward design rather than the training data. This is a meaningful conceptual shift from prior easy-to-hard data curricula, and the paper explains this framing clearly. The idea of stratifying rubrics by empirical learnability and scheduling them over training is sensible and reasonably motivated.

2 The method is easy to follow. The two-phase structure (1) generalized rubric construction/stratification and (2) dynamic curriculum learning is clearly described. The rubric table and the curriculum formulation make the framework understandable, and the paper is generally well organized.

3 The experiments are reasonably comprehensive. The authors evaluate on seven benchmarks spanning mathematical and general visual reasoning, compare against several open-source reasoning models, and include ablations on aggregation/scheduling, reward balance α, and sliding-window size.

weakness

Although the paper argues that generalized rubrics are cheaper than instance-specific rubric generation, the training pipeline still depends on a powerful teacher model for rubric candidate generation and a very large judge model for reward evaluation. This raises practical concerns about computational cost, reproducibility, and accessibility for broader adoption. The method may be much harder to use in practice than the paper’s relatively clean formulation suggests.

---

> ### Author Rebuttal · Authors · 2026-03-31
>
> We thank the reviewer for the constructive feedback and for recognizing the conceptual novelty and clear presentation of our reward-centric curriculum learning approach. We also appreciate the important concern regarding computational cost, reproducibility, and accessibility.
> >W1: Computational cost.
>
> We agree that the judge model remains a nontrivial cost. Our choice of a strong judge is mainly intended to ensure stable and informative reward evaluation, enabling fine-grained supervision of intermediate reasoning steps rather than relying solely on final-answer correctness, which is known to induce reward hacking and spurious reasoning.
>
> * Our main clarification is that RuCL is designed to balance this improved supervision with computational efficiency. Compared to prior instance-specific rubric pipelines[1,2,3], RuCL replaces per-instance rubric generation with a shared generalized rubric pool, reducing rubric generation cost from $\mathcal{O}(N\times C_{gen})$ to $\mathcal{O}(1\times C_{gen})$ with respect to the number of unique training queries.
> * Importantly, even when employing a strong reward model (RM) or judge, the additional cost is primarily incurred during offline training and does not significantly impact deployment-time inference cost. In contrast, instance-specific rubric pipelines require continuous rubric generation and evaluation for each new input, leading to persistent overhead during both training and deployment.
>
> Therefore, RuCL provides a more favorable **trade-off** between reward quality and computational cost: it improves the richness and reliability of supervision while maintaining better scalability and lower amortized cost in practical settings. We will further clarify this trade-off and provide additional discussion in the appendix.
>
> [1] Rubrics as Rewards: Reinforcement Learning Beyond Verifiable Domains
>
> [2] Reinforcement Learning with Rubric Anchors
>
> [3] Breaking the Exploration Bottleneck: Rubric-Scaffolded Reinforcement Learning for General LLM Reasoning
>
>
> >W2: Reproducibility and accessibility.
>
> We agree with the reviewer that reproducibility and accessibility are important considerations for rubric-based RL frameworks.
>
> * We note that Appendices D and E already include some key components of our pipeline, including the rubric construction process, representative rubric generation examples, and the prompts used for the judge model. These are intended to provide initial transparency into the design of RuCL.
> * To further improve clarity and reproducibility, we will revise the paper to more explicitly separate (i) the one-time offline cost of rubric construction and (ii) the online cost of reward evaluation.
> * Furthermore, we will open-source all generated rubrics, evaluation prompts, model weights, and the complete training codebase to ensure full reproducibility and facilitate broader adoption by the community.
>
> We hope this clarification addresses the reviewer’s concern. More importantly, RuCL shifts curriculum learning from data selection to reward design, which reduces methodological overhead and makes rubric-based RL more reusable and scalable than instance-specific rubric construction.

---

> > ### Author Rebuttal · Reviewer_CeUu · 2026-04-03
> >
> > The author has addressed my concern, I will keep my original score.

---

> > > ### Author Response · Authors · 2026-04-07
> > >
> > > We sincerely thank the reviewer for the constructive feedback and for recognizing that the concerns have been adequately addressed. We especially appreciate the insightful comments on computational cost, reproducibility, and accessibility, which have helped us improve both the clarity and practical relevance of our work. In response, we have strengthened the discussion on efficiency trade-offs, clarified the distinction between offline and online costs, and enhanced the reproducibility details. We believe these improvements make our work more transparent and easier to adopt in practice.

---

### Decision · Program_Chairs · 2026-04-30

**Decision:**

Accept (regular)

**Comment:**

After rebuttal, all reviewers vote for weak accept. On the other hand, two reviewers keep their concerns regarding several technical and writing issues. Notably, the reviewers still question about the static weighting mechanism, the danger of reward hacking, and the dependence on the judge LLMs. Overall, the major strengths of the paper lie in the overall reasonable curriculum learning idea and the significant empirical performance. While the limitations of lacking in-depth mechanism analysis remain. I would like to suggest an acceptance with reservation.